# VoG: Enhancing LLM Reasoning through Stepwise Verification on Knowledge Graphs

**Wenxin Zhao**[1], **Jiachuan Wang**[2,*] **Yongqi Zhang**[1], **Shuangyin Li**[3], **Cheng Deng**[1], **Jun Wang**[4], **Lei Chen**[1,5]

[1]The Hong Kong University of Science and Technology (Guangzhou)
[2]University of Tsukuba    [3]South China Normal University    [4]University College London
[5]The Hong Kong University of Science and Technology

## ABSTRACT

Large Language Models (LLMs) excel at various reasoning tasks but still encounter challenges such as hallucination and factual inconsistency in knowledge-intensive tasks, primarily due to a lack of external knowledge and factual verification. These challenges could be mitigated by leveraging knowledge graphs (KGs) to support more reliable LLM reasoning. However, existing KG-augmented LLM frameworks still rely on static integration mechanisms that cannot adjust reasoning in response to evolving context and retrieved evidence, resulting in error propagation and incomplete reasoning. To alleviate these issues, we propose **V**erify-**o**n-**G**raph (**VoG**), a scalable and model-agnostic framework to enhance LLM reasoning via iterative retrieval, stepwise verification, and adaptive revision. Besides performing KG retrieval guided by an initially generated reasoning plan, VoG iteratively verifies and revises the reasoning plan, correcting intermediate errors in consideration of the varying contextual conditions. During plan revision, VoG leverages a context-aware multi-armed bandit strategy, guided by reward signals that capture uncertainty and semantic consistency, to enhance the alignment between the reasoning plan and retrieved evidence in a more adaptive and reliable way. Experimental results across three benchmark datasets show that VoG consistently improves both reasoning accuracy and efficiency. Our code is available at `https://github.com/WenxinAZhao/VoG`.

## 1 INTRODUCTION

Despite the impressive reasoning capabilities across various natural language understanding and generation tasks (Guo et al., 2025; OpenAI, 2023), large language models (LLMs) continue to face challenges in solving knowledge-intensive tasks that require multi-hop reasoning (Ji et al., 2023; Bang et al., 2023). The essential limitation lies in the lack of up-to-date or specialized knowledge not included in their pre-training stage and the limited transparency and explainability in their reasoning processes. To address these issues, Knowledge Graphs (KGs) (Bollacker et al., 2008; Auer et al., 2007; Suchanek et al., 2007) have been adopted as promising external knowledge sources due to their explicit, organized and updatable nature (Agrawal et al., 2024; Wang et al., 2023a; Pan et al., 2023).

Existing frameworks generally follow two main directions to enhance LLM reasoning with KG. Previous methods enable LLMs to **plan** structured paths or logical forms, such as SPARQL (Pérez et al., 2009) queries, before interacting with KG (Luo et al., 2024a; Li et al., 2023; Luo et al., 2024b; Gu et al., 2022). While planning approaches facilitate structured inference, they often require expensive fine-tuning and are vulnerable to retrieve-related errors (e.g., non-executable queries or non-existent entities). Thus, several studies focus on optimizing the **retrieval** process to better support LLM reasoning. Typical methods including retrieving KG triplets and presenting them statically to the LLM (Zhang et al., 2024; Zhao et al., 2024; Wang et al., 2023b; Wen et al., 2024; Yasunaga et al., 2022) and conducting stepwise retrieval with LLM agent (SUN et al., 2024; Huang et al., 2024), both aiming to ground reasoning in structured knowledge of KG. Yet, these approaches still lack effective mechanisms to align retrieval with evolving subgoals, prompting recent work to integrate planning into the retrieval process to better guide multi-step inference (Jiang et al., 2024; Guan et al., 2024).

---

*Correspondence to Jiachuan Wang.

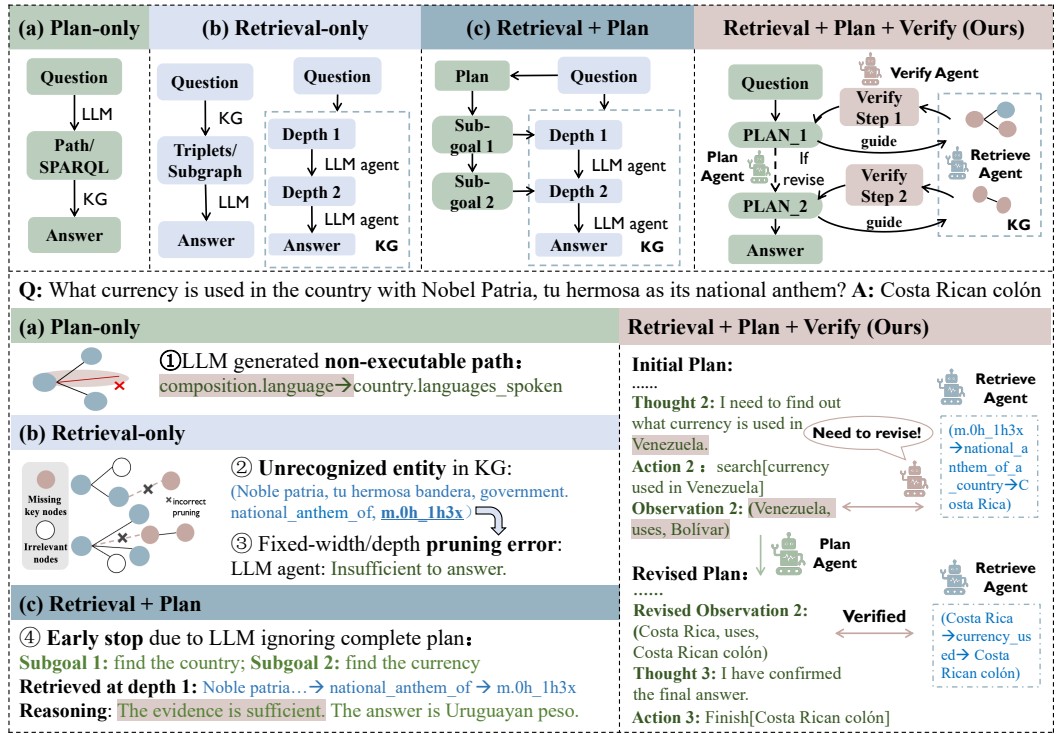

Figure 1: A comparison of existing KG-enhanced reasoning frameworks using a representative question example. Errors resulting from LLM hallucinations are highlighted in red. Text in green denotes LLM-generated content, while KG-retrieved triplets are shown in blue .

However, current KG-enhanced LLM frameworks still suffer from below challenges: (1) **Inflexible reasoning:** exiting works either follow predefined reasoning paths or conduct graph search based on fixed parameters (e.g., depth and width), as shown in Figure 1(b), leading to incomplete utilization of KG evidence and cascading reasoning errors. (2) **Limited use of information:** Most agent-based frameworks focus solely on retrieved triplets at each local step, ignoring global context such as prior reasoning steps and forward-looking relation, which makes them vulnerable to unrecognized entities at intermediate steps and prone to premature termination as illustrated in Figure 1(c).

To address these issues, we propose a novel **V**erify-**o**n-**G**raph **(VoG)** framework that supports dynamic and context-aware LLM reasoning over KG. Specifically, VoG employs a framework involving three specialized LLM agents that collaboratively perform retrieval, verification, and revision in an iterative manner. Initially, the *plan agent* generates a reasoning plan inspired by ReAct (Yao et al., 2023), which serve both as a global-level roadmap for retrieval and as contextual memory for maintaining coherence across long reasoning chains. To address the inflexible reasoning and mitigate the error propagation, VoG performs stepwise verification to detect reasoning inconsistencies as they arise and ensure the correctness of subsequent reasoning steps. Furthermore, to overcome the limitations of purely local reasoning, VoG strategically incorporates KG-grounded feedback and contextual information to revise its reasoning plan by proposing a multi-armed bandit (MAB) context selector. In summary, our main contributions are as follows:

- We propose a novel framework that enables stepwise verification on KG to mitigate error propagation during multi-hop reasoning. Through iterative refinement of reasoning plans, we process the adaptive KG retrieval to collect relevant KG feedback for the targeted question.

- We introduce a KG-aware multi-armed bandit (MAB) mechanism for adaptive context selection, which dynamically determines the maximally informative subset of KG feedback and reasoning history at each step to enhance factual consistency.

- We implement VoG and evaluate it on three KGQA datasets. Results on both open-source and closed-source LLMs validate that our framework outperforms the state-of-the-art baselines and generalizes robustly across diverse backbones.

## 2  PRELIMINARY

We introduce the preliminaries used in this paper as follows.

**Definition 1 (Knowledge Graph (KG):)** *A knowledge graph $\mathcal{G}$ is represented as a collection of factual triplets, formally defined as $\mathcal{G} = \{(e_{head}, r, e_{tail})\}$, where each triplet consists of a head entity $e_{head}$, a tail entity $e_{tail}$ and their relation $r$.*

**Definition 2 (Reasoning Plan:)** *Given a question $Q$, LLM generates a structured reasoning chain $S = [s_1, s_2, \ldots, s_T]$, in which the t-th step $s_t$ consists of the t-th thought, action, and corresponding predicted observation, denoted as $(T_t, A_t, Pred\_O_t)$.*

**Definition 3 (KG Feedback:)** *At each reasoning step t, executing retrieval guided by the action $A_t$ retrieves a set of KG triplets relevant to the current reasoning context, denoted as $O_t = [o_1^{(t)}, o_2^{(t)}, \ldots]$, where each $o = (e_{head}, r, e_{tail})$ is a triplet in $\mathcal{G}$. These triplets serve as the KG feedback at depth t to support verification and revision of the reasoning plan.*

**Problem Statement**: Given a question $Q$, a knowledge graph $\mathcal{G}$, and a set of topic entities explicitly mentioned in $Q$, the goal of multi-hop knowledge graph question answering (KGQA) is to find the answer entities are multiple hops away from the topic entities over the KG. Considering the flexibility and the training cost, we follow previous agent-based work(Chen et al., 2024; SUN et al., 2024) that iteratively retrieve and reason over KG. In this stepwise manner, we aim to guide retrieval with *reasoning plans* and use the *KG feedback* to revise the reasoning plan in turns.

## 3  METHOD

As mentioned in previous section, existing methods exhibit limited ability to adjust reasoning dynamically based on KG feedback. To address this gap, we propose **VoG**, which first generates an initial reasoning plan, retrieves supporting KG evidence for verification, and revises the plan upon detecting inconsistencies. An overview framework is given in Figure 2.

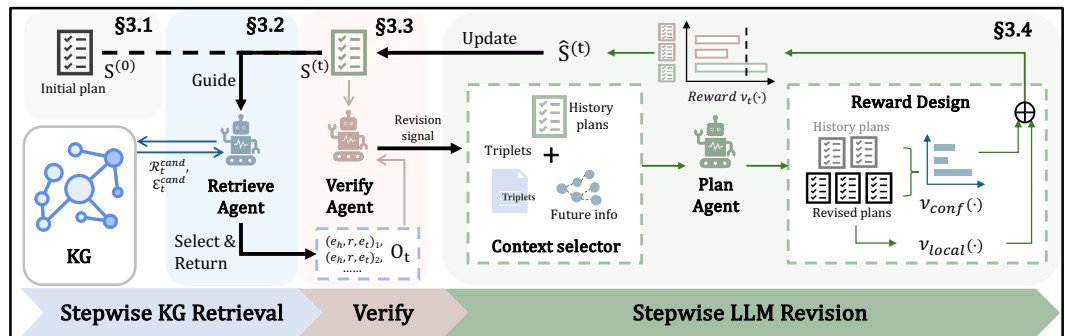

Figure 2: Overview of the VoG framework. An initial reasoning plan is first generated (§3.1) to guide the *retrieve agent* to perform stepwise retrieval (§3.2). If the *Verify agent* gives the revision signal based on retrieved feedback and plan (§3.3), the revision is conducted by the *plan agent*(§3.4).

### 3.1  INITIALIZATION AND PLANNING

Given a question $Q$, we follow mainstream work and utilize a *plan agent* to generate a complete multi-hop reasoning plan over the KG, which guide downstream retrieval and reasoning processes. The initial plan is denoted as $S^{(0)} = [s_1, s_2, \ldots, s_T]$, where $T$ is the total number of steps. Based on it, one can retrieve information from KG based on plan $S^{(0)}$ step by step. This iterative process continues until all planned steps $T$ are executed and verified. Note that $T$ implicitly defines the reasoning depth, which allows *plan agent* to adaptively adjust it dynamically based on the execution of the plan itself, rather than relying on fixed-depth settings.

## 3.2 STEPWISE KG RETRIEVAL

In this section, we design a two-stage retrieval mechanism guided by the reasoning plan.

**Plan-Guided Relation Retrieval.** At each reasoning step $t$, VoG retrieves relevant KG relations guided by the action $A_t$ from the current plan step (Figure 3). Let $\mathcal{E}_{t-1} = \{e_1^{(t-1)}, \ldots, e_N^{(t-1)}\}$ be the set of entities obtained in the previous step. We execute structured KG queries (Appendix A.1) to enumerate adjacent relations for each $e_i^{(t-1)}$, forming a candidate set $\mathcal{R}_t^{\text{cand}}$. The retrieve agent is then prompted, with the current action $A_t$ as input, to select a set of relations $\mathcal{R}_t$ that are most relevant to the reasoning objective. To mitigate noise in large candidate sets, we apply entropy-based adaptive sampling using Sentence-BERT similarity scores (Reimers and Gurevych, 2019). Filtering and prompt design details are provided in Appendix A.2 and A.3.2.

**Plan-Guided Entity Retrieval.** Based on the selected relations, VoG retrieves new entities to update KG feedback. For each selected $r \in \mathcal{R}_t$ and entity $e_i^{(t-1)}$, we query the KG using patterns $(e_i^{(t-1)}, r, ?)$ and $(?, r, e_i^{(t-1)})$, yielding a candidate entity set $\mathcal{E}_t^{\text{cand}}$. The same entropy-based sampling is applied when needed, helping the *retrieve agent* select from $e^{(t)} \in \mathcal{E}_t^{\text{cand}}$ to obtain $\mathcal{E}_t$ and append the corresponding triplets $(e_i^{(t-1)}, r, e^{(t)})$ to $O_t$, using the original question $Q$ and predicted observation $Pred\_O_t$ as context. Prompt details are in Appendix A.3.3. This process reduces irrelevant expansions and ensures that subsequent reasoning is well supported.

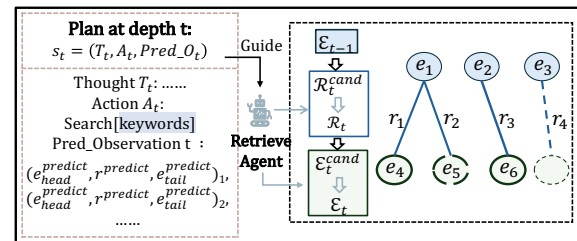

Figure 3: Plan-Guided Retrieval

## 3.3 STEPWISE KG VERIFICATION

Most of the existing works conduct retrieval on KG simply following the initial reasoning plan (Jiang et al., 2024; Chen et al., 2024) that could be non-optimal or even misleading. To mitigate error propagation and enhance factual consistency, we introduces a stepwise KG verification mechanism that integrates retrieved KG evidence as feedback.

Specifically, given $s_t$ in the generated plan $S$, VoG obtains a set of feedback $O_t = \{(e_{head}^{(t)}, r, e_{tail}^{(t)})\}$, consisting of KG triplets. Using them as factual information, a *verify agent* is first prompted to compare it against the predicted observation $Pred\_O_t \in s_t$. To further enhance the reliability, we additionally implement a pretrained DeBERTa verifier (He et al., 2021) as a secondary check. The prompts are provided in the Appendix A.3.4. Formally, we define the revision signal as:

$$\mathcal{V}(Q, s_t, O_t) = \begin{cases} 1 & \text{if } Pred\_O_t \in s_t \text{ is inconsistent with } O_t, \\ 0 & \text{otherwise.} \end{cases} \tag{1}$$

When $\mathcal{V}(Q, s_t, O_t) = 1$, VoG flags the current step as unreliable and triggers a revision process. This mechanism ensures alignment between the reasoning and retrieved KG feedback, allowing the agent to iteratively refine its plan before proceeding to the next step. Once we obtain a positive revision signal from above verification, we can handle the inconsistency issue by revising the reasoning plan. A detailed analysis of revision signal triggering is provided in Appendix E.1.

## 3.4 STEPWISE PLAN REVISION

Effective revision after verification hinges on addressing two central challenges. (i) The inherent sensitivity of LLMs to contextual inputs, together with the varying contextual requirements across different scenarios, necessitate a more dynamic strategy that can explore diverse context configurations while exploiting the most effective ones to support accurate revision. (ii) The potential of LLMs to generate hallucinated or redundant steps that misguide later reasoning highlights the need for evaluation, ensuring that each revision step is well aligned and verifiable against factual knowledge from KG.

To address these challenges, we propose a dynamic revision framework that couples adaptive context selection with explicit reward evaluation. At each revision step $t$, the *plan agent* proposes a revised plan $S^{(t)}$ with selected context, which is executed from $t+1$ onward if meeting the reward criterion. In the following, we detail how VoG achieves stepwise revision by flexibly adjusting context use to tackle (i) while ensuring conciseness and factual consistency with KG feedback to address (ii).

### 3.4.1 CONTEXT-AWARE UCB SCORING

The sensitivity of LLMs to contextual inputs implies that static heuristics or fixed strategies are brittle. To adaptively select the best contextual scope at each step, we formulate context selection as a multi-armed bandit (MAB) problem. As shown in Sec 4.5, the high context-dependence of strategy effectiveness further underscores the necessity of an adaptive selector.

Specifically, we define three complementary strategies $\mathcal{C} = \{Local, Lookahead, Global\}$ corresponding to minimal, proactive, and comprehensive use of context. These capture the main modes of leveraging KG feedback and reasoning history, and thus serve as the candidate arms in our selector:

- ***Local*** that focuses solely on the immediate KG feedback $O_t$, resulting in the input form $f_{\text{revise}}(S^{(t)}, O_t)$, which can effectively correct hallucinations when explicit triples are available.
- ***Lookahead*** that further incorporates future relations $R_{t+1}$ to the input $f_{\text{revise}}(S^{(t)}, O_t, R_{t+1})$, enabling proactive adjustment to avoid incomplete reasoning.
- ***Global*** which aggregates the full reasoning plan and all past KG feedback $O_{1:t}$, leading to the input $f_{\text{revise}}(S^{(t)}, O_{1:t})$. This strategy enables broader reassessment, which is particularly useful when accumulated errors or query intent drift occur.

To select among these strategies, we adopt the classical Upper Confidence Bound (UCB) algorithm (Kaufmann et al., 2012) and further extend it with context-aware priors. Formally, for each candidate context strategy $c \in \mathcal{C}$, we record the number of times it has been selected $N_c$ and its cumulative reward $R_c$. At each step $t$, the UCB score is then defined as follows, with the robustness of this design further validated through ablation and sensitivity analyses in Appendix E.4 and E.5.

$$\text{UCB}_t(c) = \underbrace{\frac{R_c}{N_c}}_{\text{Exploitation}} + \underbrace{\alpha\sqrt{\frac{\log N}{N_c}}}_{\text{Exploration}} + \underbrace{\mathcal{B}_{\text{ent}}(H_t) + \mathcal{B}_{\text{KG}}(t, E_{\text{rep}}) + \mathcal{B}_{\text{div}}(c)}_{\text{Context-aware Priors}}, \tag{2}$$

where $\alpha$ is a fixed constant and $N$ is the number of total selection times. Here, $H_t$ denotes the normalized entropy of the current answer distribution, which signals the level of uncertainty like in previous work (Kuhn et al., 2023), and $E_{\text{rep}}$ denotes the set of entities repeatedly retrieved from the accumulated KG feedback $O_{1:t}$. We then incorporate three context-aware bonus terms: an entropy-based bonus $\mathcal{B}_{\text{ent}}(H_t)$ that promotes exploration under high answer uncertainty, a KG-aware bonus $\mathcal{B}_{\text{KG}}(t, E_{\text{rep}})$ that penalizes repetitive retrieval and a diversity bonus $\mathcal{B}_{\text{div}}(c)$ that discourages repeated selection of the same strategy. Precise definitions for each term and the full algorithm are detailed in Appendix B.1.

### 3.4.2 REWARD DESIGN

To guide strategy selection, we assess the effectiveness of the chosen context strategy via the quality of its revised plan. Once a strategy $c_t \in \mathcal{C}$ is selected, the *plan agent* generates a candidate reasoning plan. Considering the inherent unreliability and instability of LLM outputs, we design a scalar reward $\nu_t$ that jointly captures step-level coherence and global answer stability as detailed below.

**Task-specific Reward.** The task-specific reward $\nu_{\text{local}}$ measures the local quality of a revision through heuristic signals derived from the reasoning process. The stepwise nature of VoG reveals a heterogeneous set of contextual evidence at each reasoning step, including KG feedback, historical reasoning steps, and the evolving reasoning plan that reflects the model's interpretation of the question. This enables us to evaluates revisions from a broader range of perspectives, reducing the risk of any potentially biased single signal dominating the revision process. We incorporates five key metrics to evaluate the quality of each revision as shown below:

- **Validation:** Verifies the revision by assessing the consistency of revised observation $Pred\_O_t$ against KG feedback, which ensures that the revision is validated by factual knowledge in KG.

- **Quality:** Evaluates the semantic relevance between the revised observation $Pred\_O_t$ and the question $Q$. This metric assesses the quality of revision process by ensuring that the revision aligns with the original query intent.
- **Question Alignment:** Ensures that the reasoning process maintains a consistent goal throughout multi-hop reasoning by measuring the similarity between reasoning sub-steps and the query.
- **Thought Coherence:** Ensures the alignment between the current reasoning step $T_t$ and the relevant KG feedback, verifying that the reasoning is coherent with the knowledge provided.
- **Efficiency:** Penalizes reasoning steps that repeat or overlap significantly, reducing redundant reasoning and encouraging more efficient and progressive paths.

In contrast to prior work that relies on LLM-based scoring (Sui et al., 2025; Zheng et al., 2023), we adopt lightweight pre-trained models for these checks, which makes the evaluation both efficient and scalable without incurring additional inference overhead. Full details are provided in Appendix B.2.

**Confidence-Based Reward.** The confidence-based reward $\nu_{\text{conf}}$ measures the stability of the final answer across candidate reasoning plans generated during revision. Formally, let $\mathcal{P} = \{S_1, S_2, \ldots, S_N\}$ denote the set of candidate reasoning plans generated during revision, and $a_{S_i}$ be the final answer of plan $S_i$. To measure confidence here, we use subscript $i$ exclusively to index all candidate plans in $\mathcal{P}$ regardless of step. We use $\sim$ to denote semantic equivalence between answers and define the reward $\nu_{\text{conf}}$ for a candidate plan $S_i$ as:

$$\nu_{\text{conf}}(S_i)) = \frac{1}{|\mathcal{P}|} \sum_{S \in \mathcal{P}} \mathbb{I}(a_S \sim a_{S_i}). \tag{3}$$

This reward encourages convergence toward stable, high-confidence answers that are consistently supported across multiple candidate plans.

**Entropy-aware Integration.** To adaptively balance local and global signals, we introduce an entropy-aware weighting scheme. Let $H_t \in [0, 1]$ denote the normalized entropy of the answer distribution at step $t$, and $\beta \in (0, 1)$ a scaling factor. We compute:

$$\lambda_t = \beta \cdot \exp(-H_t), \quad \nu_t = (1 - \lambda_t)\,\nu_{\text{local}} + \lambda_t\,\nu_{\text{conf}}. \tag{4}$$

Intuitively, when the answer distribution is uncertain (high $H_t$), the global consensus reward dominates; when the distribution is confident, local reasoning quality takes precedence. The resulting scalar reward is then accumulated as $R_c$ to update the UCB score in Eq. (2), thus closing the loop between revision evaluation and adaptive strategy selection. Ablation on the reward design is provided in Appendix E.6.

## 4 EXPERIMENT

### 4.1 EXPERIMENT SETUP

**Datasets.** To evaluate the effectiveness of VoG in enhancing the reasoning capabilities of large language models on knowledge-intensive tasks, we conduct experiments across three widely-used datasets. Specifically, we conduct experiments on two multi-hop KGQA datasets, ComplexWebQuestions (CWQ) (Talmor and Berant, 2018) and WebQuestionsSP (WebQSP) (Yih et al., 2016), as well as the open-domain QA dataset WebQuestions (Berant et al., 2013). All datasets rely on Freebase (Bollacker et al., 2008) as the underlying factual source. Following previous studies(SUN et al., 2024; Chen et al., 2024), we adopt exact match accuracy (Hits@1) as our primary evaluation metric and also report F1 scores for completeness. We show the details of above datasets in Appendix C.

**Baselines.** To comprehensively evaluate our framework, we compare VoG against three major categories of baseline approaches: (i) *LLM-only methods*, (ii) *fine-tuned methods*, and (iii) *LLM agent+KG methods*. For *LLM-only* baselines, we adopt IO prompting (Brown et al., 2020) and CoT (Trivedi et al., 2023). For *fine-tuned methods*, we include KD-CoT (Wang et al., 2023b), DECAF (Yu et al., 2023), RoG (Luo et al., 2024b), UniKGQA (Jiang et al., 2022) and GNN-RAG (Mavromatis and Karypis, 2024). In addition, we further include methods that fine-tune only the retriever while keeping the LLM frozen, such as SubgraphRAG (Li et al., 2025) and the unfine-tuned LLM variant of GNN-RAG (Mavromatis and Karypis, 2024). For *LLM agent-based* approaches, we compare with ToG (SUN et al., 2024) and PoG (Chen et al., 2024). Full implementation details of all baselines are provided in the Appendix D.

## 4.2  MAIN RESULTS

Table 1 summarizes the performance of VoG across three benchmark datasets compared to representative state-of-the-art (SOTA) baselines. Overall, VoG outperforms all included baselines across all categories of methods.

First, compared to *LLM-only* baselines, VoG significantly improves performance by incorporating structured knowledge from external KGs. This highlights the importance of factual grounding in multi-hop reasoning, particularly for complex questions where parametric knowledge alone is insufficient. Second, we compare VoG to *fine-tuned methods*. Despite not requiring additional fine-tuning, VoG achieves competitive or superior performance, demonstrating the effectiveness of its verification-driven and plan-adaptive design.

Finally, when compared to *agent-based reasoning frameworks*, VoG attains higher accuracy through stepwise verification and adaptive context selection mechanisms. To assess generalizability and effectiveness across different model sizes, we further evaluate VoG using smaller-scale LLMs (e.g., Qwen2.5-7B (Team, 2024)). Even under reduced model capacity, VoG achieves robust improvements over baseline agents, confirming that its gains stem from methodological advances rather than reliance on model scale. Beyond our main experiment conducted on Freebase (Bollacker et al., 2008), we further demonstrate VoG's generalization ability on Wikidata (Vrandečić and Krötzsch, 2014), as shown in Appendix G.

Table 1: Performance comparison of different methods across datasets. Bold indicates the best agent performance for each backbone.

| Method | CWQ | | WebQSP | | WebQuestions | |
|---|---|---|---|---|---|---|
| | EM | F1 | EM | F1 | EM | F1 |
| *LLM-only* | | | | | | |
| *GPT-3.5* | | | | | | |
| IO prompt (Brown et al., 2020) | 37.6 | - | 63.3 | - | 48.7 | - |
| CoT (Trivedi et al., 2023) | 38.8 | - | 62.2 | - | 48.5 | - |
| SC (Wang et al., 2022) | 45.4 | - | 61.1 | - | 50.3 | - |
| *Fine-tuned methods* | | | | | | |
| *Fine-tuned LLM* | | | | | | |
| KD-CoT (Wang et al., 2023b) | 55.7 | - | 68.6 | 52.5 | - | - |
| UniKGQA (Jiang et al., 2022) | 51.2 | 48.0 | 77.2 | 70.2 | - | - |
| RoG (Luo et al., 2024b) | 62.6 | 56.2 | 80.4 | 70.8 | - | - |
| DECAF (Yu et al., 2023) | 70.4 | - | 82.1 | - | - | - |
| KG-Agent (Jiang et al., 2024) | 72.2 | - | 83.3 | - | - | - |
| GNN-RAG+RA (Mavromatis and Karypis, 2024) | 68.7 | 60.4 | 90.7 | 73.5 | – | – |
| *Fine-tuned Retriever + GPT-3.5* | | | | | | |
| GNN-RAG (Mavromatis and Karypis, 2024) | 64.1 | - | 85.3 | - | – | – |
| SubgraphRAG (Li et al., 2025) | 56.3 | 49.1 | 83.1 | 69.2 | – | – |
| *LLM Agent + KG* | | | | | | |
| *Qwen2.5-7B* | | | | | | |
| ToG (SUN et al., 2024) | 42.5 | 28.7 | 56.0 | 37.3 | 39.9 | 31.3 |
| PoG (Chen et al., 2024) | 46.0 | 31.4 | 58.5 | 40.4 | 46.2 | 30.3 |
| **VoG (Ours)** | **53.3** | **45.6** | **67.3** | **55.1** | **52.8** | **45.2** |
| *GPT-3.5* | | | | | | |
| ToG (SUN et al., 2024) | 58.9 | 41.9 | 76.2 | 50.9 | 54.5 | 39.3 |
| PoG (Chen et al., 2024) | 63.2 | 43.7 | 82.0 | 58.1 | 61.7 | 44.3 |
| **VoG (Ours)** | **64.7** | **56.2** | **83.2** | **69.1** | **63.0** | **61.3** |
| *GPT-4* | | | | | | |
| ToG (SUN et al., 2024) | 67.6 | 47.6 | 82.6 | 58.9 | 57.9 | 44.9 |
| PoG (Chen et al., 2024) | 75.0 | 42.1 | 87.3 | 59.8 | 71.7 | 44.5 |
| **VoG (Ours)** | **77.6** | **67.5** | **88.7** | **73.2** | **72.3** | **61.7** |

## 4.3 ABLATION STUDY

We conduct the ablation studies on CWQ and WebQSP using GPT-3.5 as the backbone model to provide a comprehensive view of how VoG achieves its performance gains.

**Impact of the verification and adaptive revision.** We first examine the stepwise verification mechanism and the contribution of context selector of VoG as shown in Table 2. In the *w/o Context Selector* variant, the bandit-based adaptive strategy is replaced with a single fixed strategy, so revisions are performed without dynamic strategy at each step. In the *w/o Verify+Revise* variant, the *plan agent* directly outputs the answer from its initial plan without performing retrieval, verification, or revision as the *plan-retrieve-revise* process in VoG constitutes a unified feedback loop. Interestingly, this plan-only configuration still outperforms standard CoT and Self-Consistency baselines, suggesting that LLM-internal planning alone offers strong multi-hop reasoning capabilities, but remains vulnerable to error propagation without external verification and revision.

Table 2: Ablation Study on VoG's Stepwise Verification and Adaptive Revision.

| Variant | CWQ | WebQSP | WebQuestions |
|---|---|---|---|
| **VoG** | **64.7** | **83.2** | **63.0** |
| w/o Context Selector | | | |
| *Local* | 60.1 ($\downarrow$4.6) | 80.6 ($\downarrow$2.6) | 58.2 ($\downarrow$4.8) |
| *Lookahead* | 63.6 ($\downarrow$1.1) | 81.4 ($\downarrow$1.8) | 59.9 ($\downarrow$3.1) |
| *Global* | 60.2 ($\downarrow$4.5) | 80.3 ($\downarrow$2.9) | 60.6 ($\downarrow$2.4) |
| w/o Verify+Revise | 51.7 ($\downarrow$13.0) | 72.1 ($\downarrow$11.1) | 55.8 ($\downarrow$7.2) |

**Effect of Different Context Strategies.** We further analyze the effectiveness of three fixed strategies and compare them with our KG-aware context selector. Each variant in the *w/o Context Selector* setting above corresponds to a fixed strategy applied end-to-end throughout the reasoning process. All fixed strategies fall short, since none can adapt effectively across diverse reasoning scenarios. In contrast, our method adaptively selects the most suitable context at each step to achieve more accurate and flexible reasoning. A detailed analysis of strategy-level performance, including accuracy and revision rates per iteration, is provided in Appendix E.3.

## 4.4 EFFICIENCY ANALYSIS

We further assess the computational efficiency of VoG by the average token consumption per query, as conducted in prior agent-based baselines. As shown in Table 3, VoG significantly reduces token usage and maintain comparable or lower levels of LLM interaction. A detailed token breakdown across planning, retrieval, verification, and revision steps is provided in Appendix E.2. We also analyze the trade-offs between complexity and performance gains in Appendix E.7.

Table 3: Efficiency analysis on average tokens and calls per query compared with existing methods.

| Method | CWQ | | WebQSP | |
|---|---|---|---|---|
| | Tokens | Calls | Tokens | Calls |
| ToG | 9669.4 | 22.6 | 6031.2 | 15.9 |
| PoG | 8156.2 | 13.3 | 5517.7 | 9.0 |
| **VoG** | 6566.8 | 14.5 | 3439.0 | 9.4 |

This efficiency gain stems from two key aspects. From a methodological perspective, VoG adopts an adaptive retrieval strategy to prioritizes relevant information and employs a context selector that contributes to reducing unnecessary input. Moreover, its plan-guided retrieval enables adaptive control over depth and breadth, avoiding the irrelevant exploration introduced by beam search in ToG and thus improving overall retrieval efficiency. From an implementation standpoint, VoG stores context implicitly within the reasoning plan, avoiding external memory modules as required in PoG and enabling direct answer extraction without additional LLM calls. Together, these design choices make VoG a more practical and lightweight solution for scalable KG reasoning, especially under limited computational budgets.

## 4.5 CASE STUDY

In this section, we present a fine-grained case in Figure 4 using Qwen2.5-7B and visualized attention heatmaps. In the presented case, the *Local* strategy, which is typically effective when explicit triples are available, becomes limited here because the entity exists in the KG but lacks a resolvable name. By contrast, the *Lookahead* strategy, which observes future relations, and the *Global* strategy, which

helps avoid query intent drift, both succeed in revising the plan. However, providing all contextual information at once distracts the LLM and induces hallucinations. This case demonstrates how adaptively selecting complementary contexts can enhance plan refinement in unpredictable scenarios, such as incomplete KG signals or LLM hallucinations, without relying on manually predefined rules. Additional cases demonstrating how different strategies perform under varying conditions, along with comparisons to ToG and PoG and an analysis of recovered failure types, are provided in Appendix F.

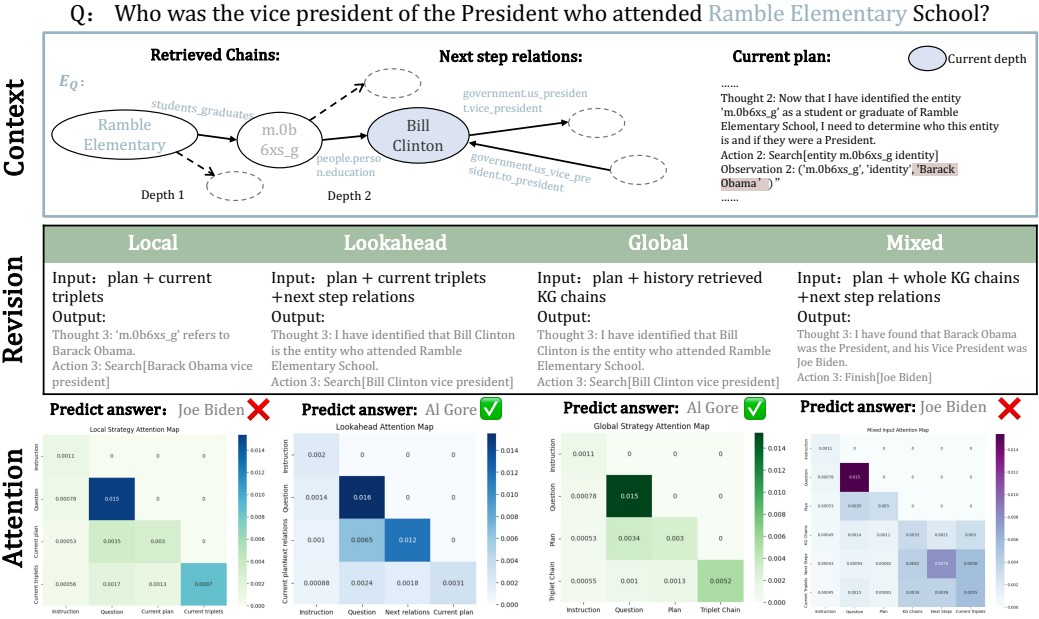

Figure 4: Case comparison of revision behavior across context strategies and an mixed input.

## 5 RELATED WORK

### 5.1 LLM-BASED AGENTS

LLM-based agents have become a prominent paradigm for reasoning and decision-making, framing complex tasks as sequential interactions between planning, observation, and action. Early approaches such as ReAct (Yao et al., 2023) integrate chain-of-thought reasoning with tool use, enabling stepwise interaction with external environments. Subsequent frameworks extend this paradigm to various domains, such as web-based QA (Nakano et al., 2021), multimodal reasoning (Yang et al., 2023), social simulation (Park et al., 2023) and maths problem (Lei et al., 2024; Chen et al., 2025). However, these approaches lack mechanisms to learn from feedback and remain heavily dependent on the underlying LLM backbone. Therefore, researchers have incorporates self-evaluated feedback (Shinn et al., 2023; Yao et al., 2023; Madaan et al., 2023; Panickssery et al., 2024) and external retrieval feedback (Jin et al., 2025) into the reasoning loop to enhance LLM reasoning. In parallel, Monte Carlo Tree Search (MCTS)-style agents have been introduced to sample and select from multiple reasoning trajectories for robust decision-making (Hao et al., 2023; Hu et al., 2025; Luo et al., 2025; Sun et al., 2025). However, relying on unverified external feedback often introduces noise and factual gaps, increasing the risk of hallucinations during reasoning.

### 5.2 KG-ENHANCED LLM REASONING

To enhance the reliability of LLM reasoning, recent studies have incorporated KGs as structured external sources. We summarize representative KG-enhanced approaches in Table 4, which broadly focus on three key aspects: planning, retrieval, and verification. Planning-focused methods such as RoG (Luo et al., 2024b) and KG-Agent (Jiang et al., 2024) enhances the planning capabilities of LLMs by generating structured reasoning paths over KGs. While effective initially, such plans are

fixed once generated and cannot adapt to new evidence, leading to error accumulation in multi-hop inference. Retrieval-focused methods aim to improve the quality of KG evidence provided to the LLM. For example, KnowGPT (Zhang et al., 2024) applies reinforcement learning for knowledge extraction, KG-CoT (Zhao et al., 2024) retrieves high-quality multi-hop subgraphs with a graph reasoning model, and ToG (SUN et al., 2024) adopts an LLM agent to perform iterative retrieval. Building upon these, PoG (Chen et al., 2024) incorporates planning with iterative retrieval to better coordinate evidence gathering and reasoning.

Despite these advances, the above methods either lack explicit verification mechanisms or perform only global verification after completing the entire reasoning process. To address this gap, KD-CoT (Wang et al., 2023b) introduces a retriever-reader-verifier pipeline that verifies the factual consistency of final answers against KG evidence. While promising, their verification serves only as a passive check, providing a

Table 4: Comparison of recent KG-enhanced LLM reasoning methods from three perspectives.

| Method | Plan | Retrieve | Verify |
|---|---|---|---|
| RoG (Luo et al., 2024b) | ✓ | ✗ | ✗ |
| KnowGPT (Zhang et al., 2024) | ✗ | ✓ | ✗ |
| KG-Agent (Zhao et al., 2024) | ✓ | ✓ | ✗ |
| KD-COT (Wang et al., 2023b) | ✗ | ✓ | ✓ |
| ToG (SUN et al., 2024) | ✗ | ✓ | ✗ |
| PoG (Chen et al., 2024) | ✓ | ✓ | ✗ |
| **VoG (Ours)** | ✓ | ✓ | ✓ |

final validation that does not influence or revise subsequent reasoning steps. As a result, errors in earlier steps can propagate without correction. While PoG (Chen et al., 2024) introduces correction at the stepwise level, its focus remains on refining local retrieval rather than the entire trajectory. Without incorporating explicit stepwise adjustment, these methods overlook the assurance of the correctness and coherence of the full reasoning process.

## 6 CONCLUSION

In this paper, we propose **Verify-on-Graph (VoG)**, a unified framework that advances trustworthy LLM reasoning by by coupling planning, retrieval, and verification into a closed-loop process over KGs. Specifically, VoG treats LLM-generated reasoning plans as tentative and refines them by detecting and correcting potential hallucinations that arise during multi-hop reasoning. Unlike prior frameworks that lack intermediate factual checking and adaptive revision, VoG enables stepwise verification and broader context integration at each step. Our experiments demonstrate that VoG significantly improves reasoning accuracy, robustness, and efficiency across multiple benchmarks, and generalizes well across LLM backbones without requiring additional training.

## ETHICS STATEMENT

The research conducted in this paper adheres to the ICLR Code of Ethics in every respect. Our study focuses on enhancing the reasoning reliability of large language models by incorporating stepwise verification with knowledge graphs. As the framework does not involve human subjects, private data, or domain-specific sensitive material, it raises no immediate concerns regarding privacy, safety, or security. All experiments are conducted on publicly available benchmarks, which contain no personal or sensitive information, and we strictly comply with their licenses and intended use.

## REPRODUCIBILITY STATEMENT

The paper fully discloses all the information needed to reproduce the main experimental results of the paper to the extent that it affects the main claims and conclusions. We provide our experimental results in Section 4, and an anonymous code repository with detailed instructions for reproducing them is referenced in the main text and also included in the supplementary materials. Detailed implementation notes and access to the models used in our framework are given in Appendix B.3. All datasets used in experiments are standard public benchmarks, with details described in Appendix C. As our framework is training-free, no model checkpoint is required, and reproducibility is ensured through the provided code, dataset references, and documented procedures.

## ACKNOWLEDGMENTS

Lei Chen's work is supported by National Key Research and Development Program of China Grant No. 2023YFF0725100, National Science Foundation of China under Grant No. U22B2060, Guangdong-Hong Kong Technology Innovation Joint Funding Scheme Project No. 2024A0505040012, AOE Project AoE/E-603/18, Theme-based project TRS T41-603/20R, CRF Project C2004-21G, Key Areas Special Project of Guangdong Provincial Universities 2024ZDZX1006, Guangdong Province Science and Technology Plan Project 2023A0505030011, HKUST(GZ) CMCC(Guangzhou Branch) Metaverse Joint Innovation Lab under Grant No. P00659, Hong Kong ITC TC-SKLCRCC26EG01, ITF grant PRP/004/22FX, Zhujiang scholar program 2021JC02X170, HKUST Webank joint research lab. Jiachuan Wang's work is supported in part by JST CREST (JPMJCR22M2). Yongqi Zhang's work is supported by Guangdong Basic and Applied Basic Research Foundation 2025A1515010304, Guangdong Province Project 2024QN11X088, Guangzhou Science and Technology Planning Project 2025A03J4491.

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

## APPENDIX

This appendix provides additional implementation details, experimental configurations, full prompt templates, ablation settings, and supplementary visualizations referenced throughout the main paper.

## A  FURTHER DETAILS OF VOG AGENTS

### A.1  PRE-DEFINED SPARQL QUERY

For KG-based knowledge extraction, we use the following predefined SPARQL query templates to access facts from Freebase.

---

**SPARQL: Relation Retrieval**

**Outgoing Relations:**

```
PREFIX ns: <http://rdf.freebase.com/ns/>
SELECT DISTINCT ?relation
WHERE {
  ns:mid ?relation ?x .
}
```

**Incoming Relations:**

```
PREFIX ns: <http://rdf.freebase.com/ns/>
SELECT DISTINCT ?relation
WHERE {
  ?x ?relation ns:mid .
}
```

---

**SPARQL: Entity Search**

$(e_{head}, r, ?)$

```
PREFIX ns: <http://rdf.freebase.com/ns/>
SELECT ?tailEntity
WHERE {
  ns:mid ns:relation ?tailEntity .
}
```

$(?, r, e_{tail})$

```
PREFIX ns: <http://rdf.freebase.com/ns/>
SELECT ?tailEntity
WHERE {
  ?tailEntity ns:relation ns:mid .
}
```

---

**SPARQL: Entity Name Search**

```
PREFIX ns: <http://rdf.freebase.com/ns/>
SELECT DISTINCT ?tailEntity
WHERE {
  {
    ?entity ns:type.object.name ?tailEntity .
    FILTER(?entity = ns:mid)
  }
  UNION
  {
    ?entity <http://www.w3.org/2002/07/owl|\#|sameAs> ?tailEntity .
    FILTER(?entity = ns:mid)
```

```
        }
    }
```

At each reasoning depth $t$, VoG constructs structured KG queries based on the current action $A_t$ from the reasoning plan $S^{(t)}$. These queries follow the pattern $(e_{head}, r, ?)$ or $(?, r, e_{tail})$ and are executed against the underlying KG. Entities from $\mathcal{E}_{t-1}$ and filtered relations from $\mathcal{R}_t^{\text{cand}}$ are combined to form the query set.

## A.2 ENTROPY-BASED FILTERING

To reduce prompt length and eliminate noise from large candidate sets, we apply entropy-based filtering to relation and entity candidates. Specifically, we first compute similarity scores between all candidates and a set of suggested relations using a pre-trained encoder `msmarco-bert-base-dot-v5`. For each candidate $c_i \in \mathcal{C}$, its weight is calculated as

$$w_i = \max_{r_j \in R} \texttt{sim}(M(c_i), M(r_j)), \tag{5}$$

where $M(\cdot)$ denotes the encoding function and $\texttt{sim}(\cdot, \cdot)$ the similarity measure.

We then normalize the weights via softmax:

$$p_i = \frac{\exp(w_i)}{\sum_j \exp(w_j)}, \tag{6}$$

and compute the normalized entropy over the distribution $\{p_i\}$:

$$H_{\text{norm}} = -\frac{\sum_i p_i \log(p_i)}{\log |\mathcal{C}|}. \tag{7}$$

Using $H_{\text{norm}}$, we adaptively determine the selection width $k$ within bounds $[k_{\min}, k_{\max}]$ by

$$k = \max\left(k_{\min}, \min\left(\lfloor k_{\min} + (k_{\max} - k_{\min}) \cdot H_{\text{norm}} \rfloor, |\mathcal{C}|\right)\right). \tag{8}$$

Finally, we select the top-$k$ candidates ranked by their weights $w_i$. The detailed procedure is summarized in Algorithm 1, where the above equations correspond to the respective steps of computing weights, normalizing via softmax, calculating entropy, determining adaptive width, and selecting candidates.

---

**Algorithm 1:** Entropy-Based Relation (or Entity) Filtering

**Input:** Candidate set $\mathcal{C} = \{c_1, \dots, c_n\}$, Suggested relations $R = \{r_1, \dots, r_m\}$, Similarity encoder $M$, Width bounds $[k_{\min}, k_{\max}]$

**Output:** Filtered subset $\mathcal{C}_{\text{selected}}$

1 Encode all candidates and suggestions using encoder $M$;
2 Compute similarity scores $w_i$ as in Equation equation 5;
3 Normalize weights via softmax according to Equation equation 6;
4 Compute normalized entropy $H_{\text{norm}}$ as per Equation equation 7;
5 Compute adaptive width $k$ following Equation equation 8;
6 Select top-$k$ candidates $\mathcal{C}_{\text{selected}}$ by weights $w_i$;
7 **return** $\mathcal{C}_{\text{selected}}$

---

## A.3 Prompt Templates

### A.3.1 Plan generation Prompt

> **Prompt Template: Plan Generation**
>
> **Instruction:** You are an intelligent assistant tasked with answering the following question. Your job is to understand the question and plan all the necessary steps to solve it. Do not judge the question or give an unknown answer. You can only use the following two actions:
> (1) Search[Keyword]: To retrieve relative information based on the given question.
> (2) Finish[Answer]: When the observations are sufficient to answer the question, return the final answer and finish the task.
>
> *Few-shot examples*
>
> **Inputs:**
> **Question**: $\{ Q \}$
> **Output:** Initial plan $S^{(t)}$

### A.3.2 Relation Selection Prompt

> **Prompt Template: Relation Selection**
>
> **Instruction:** Please provide the relevant relations to the question "`{Q}`" and suggested relation in current action "`{A_t}`".
> **Candidate Relations:** $\{r_1, r_2, ..., r_n\}$
>
> *Few-shot examples*
>
> **Output:** `{Relevant relations}`

### A.3.3 Entity Selection Prompt

> **Prompt Template: Entity Selection**
>
> **Instruction:** Based on the question "`{Q}`" and predicted observation "`{O_t}`", choose the most plausible target entities.
> **Candidate Entities:** $\{e_1, e_2, ..., e_n\}$
>
> *Few-shot examples*
>
> **Output:** `{Entities}`

### A.3.4 Verification Prompt

> **Prompt Template: Stepwise Verification**
>
> **Instructions:** You are given a set of knowledge triplets and an LLM-generated reasoning step. Analyze whether it is necessary to revise the LLM's observation. Your response must be in valid JSON format including keys `"Revise"` and `"Reason"`. If `"Revise"` is "Yes", include a corrected `"Revised Observation"` field.
> **Predicted observation from LLM:** {}
> **Knowledge Triplets:** []
> *Few-shot examples*
>
> **Output Format (JSON):**
>
> {

```
  "Revise": "Yes" or "No",
  "Reason": "...",
  "Revised Observation": (only if "Yes")
}
```

### A.3.5 REVISION PROMPT

**Prompt Template: Local Revision**

**Instruction:** You are provided with a reasoning plan for the following question. Based on the given context, revise the plan as needed to correctly answer the question. You can also adjust previous steps based on how the observation aligns with or contradicts existing steps.

*Few-shot examples*

**Inputs:**
**Question**: { $Q$ }
**Current plans**: { $S^{(t-1)}$ }
**Current observation**:{ $O_t$ }

**Output:**Revised plan $\hat{S}_t$

**Prompt Template: Global Revision**

**Instruction:** You are provided with a reasoning plan for the following question and a set of knowledge graph (KG) triplets. The existing reasoning plan might have factual errors. Please revise the reasoning process completely from Thought 1.

*Few-shot examples*

**Inputs:**
**Question**: { $Q$ }
**Current plans**: { $S^{(t-1)}$ }
**KG triplets**:{ $O_{1:t}$ }

**Output:**Revised plan $S^{(t)}$

**Prompt Template: Lookahead Revision**

**Instruction:** You are provided with a reasoning plan for the following question and future relations as reference. Based on the given context, revise the plan as needed to correctly answer the question.

*Few-shot examples*

**Inputs:**
**Question**: { $Q$ }
**Current plans**: { $S^{(t-1)}$ }
**Lookahead relations**:{ $R_{t+1}$ }

**Output:**Revised plan $S^{(t)}$

---

**Prompt Template: Mixed input**

**Instruction:** You are provided with a reasoning plan for the following question, a set of knowledge graph (KG) triplets, and future relations as reference. Based on the given context, revise the plan as needed to correctly answer the question.

*Few-shot examples*

**Inputs:**
**Question**: { $Q$ }
**Current plans**: { $S^{(t-1)}$ }
**Lookahead relations**:{ $R_{t+1}$ }
**KG triplets**:{ $O_{1:t}$ }

**Output:** Revised plan $S^{(t)}$

---

# B DETAILS OF CONTEXT SELECTOR

## B.1 ALGORITHM AND BONUS FORMULATION

Our selection is performed via a modified UCB algorithm that incorporates reward-driven updates and KG-aware priors, as shown in Algorithm 2. We extend the classical Upper Confidence Bound (UCB) algorithm to incorporate domain-specific priors for KG reasoning. Specifically, for each context-selection strategy $c \in \mathcal{C}$, the score at step $t$ is computed as Eq. (2) in Section 3.4, where each additional bonus term is defined below.

**Entropy-aware Bonus.** To encourage *global* planning under high answer uncertainty, we define a bonus term based on the entropy $H_t$ of the answer distribution at step $t$:

$$\mathcal{B}_{\text{ent}}(H_t) = \lambda_{\text{ent}} \cdot \left[ \mathbb{I}_{c=global} \cdot \sigma\left(a(H_t - b)\right), \right) \tag{9}$$

where $H_t$ is the normalized entropy of the current answer distribution. The sigmoid function $\sigma(x) = \frac{1}{1+e^{-x}}$ ensures a smooth transition of reward scaling. Constants $a = 6$ and $b = 0.5$ control the steepness and center of the response curve, respectively.

**KG-aware Bonus.** To prevent inefficient exploration due to over-deep search or entity redundancy, we incorporate a KG-aware penalty term that discourages excessive reasoning depth and repeated use of entities:

$$\mathcal{B}_{\text{KG}}(t, E_{\text{rep}}) = \lambda_{\text{KG}} \cdot \left[ \mathbb{I}_{c=global} \cdot (\delta \cdot E_{\text{rep}}) - \mathbb{I}_{c=lookahead} \cdot \tanh\left( \kappa \cdot \frac{t}{T_d} \right) \right], \tag{10}$$

Here, $c \in \{lookahead, global\}$ denotes the current strategy, $t$ is the current reasoning depth, and $T_d$ is the expected maximum depth. The binary variable $E_{\text{rep}} \in \{0, 1\}$ indicates whether the *retrieve agent* revisits a previously retrieved entity and results in a loop. The constants $\delta$ and $\kappa$ control the penalty strength for repeated entities and the steepness of the depth-based penalty curve, respectively. In our experiments, we set $\delta = 0.2$ and $\kappa = 4$. The indicator function $\mathbb{I}_{c=}$ activates the corresponding bonus based on the selected strategy.

**Strategy Diversity Penalty.** To avoid excessive reliance on a single context strategy, we incorporate a diversity-based penalty term that discourages repetitive selection. This is computed based on the number of times each strategy has been chosen within the past $k$ revision steps:

$$\mathcal{B}_{\text{div}}(c) = -\lambda_{\text{div}}^{(c)} \cdot \text{count}_k(c), \tag{11}$$

where $\text{count}_k(c)$ denotes the number of times strategy $c$ has been selected in the last $k$ steps. We apply a strategy-specific coefficient $\lambda_{\text{div}}^{(c)}$, where $c \in \{\text{Local}, \text{Lookahead}, \text{Global}\}$, allowing differentiated regularization strength depending on the relative risk of overuse for each strategy.

---

**Algorithm 2:** KG-Aware Context Selection via Modified UCB

---

1: **Input:** Strategy set $\mathcal{C} = \{$Local, Lookahead, Global$\}$, initial plan steps $T$, expected depth $T_d$ ;
2: **Initialize:** $N[c] \leftarrow 0$, $R[c] \leftarrow 0$, $\text{count}_k[c] \leftarrow 0$, $N \leftarrow 0$, $\mathcal{P} \leftarrow \emptyset$;
3: **for** $t = 1$ **to** $T$ **do**
4:   Compute entropy $H_t$ and normalized depth $d_t = t/T_d$;
5:   **while** $\nu_t < \text{reward\_threshold}$ **do**
6:    **foreach** $c \in \mathcal{C}$ **do**
7:     **if** $N[c] = 0$ **then**
8:      $\text{Score}[c] \leftarrow +\infty$ ;      // Ensure initial exploration
9:      **continue**
10:     **end**
11:     Compute $\text{UCB}_t(c)$ using Eq. 2;
    // Context-aware bonus terms:
    //   $\mathcal{B}_{\text{ent}}(H_t)$ from Eq. 9
    //   $\mathcal{B}_{\text{KG}}(t, \mathcal{E}_{\text{rep}})$ from Eq. 10
    //   $\mathcal{B}_{\text{div}}(c)$ from Eq. 11
12:     $\text{Score}[c] \leftarrow \text{UCB}_t(c)$;
13:    **end**
14:    $c_t \leftarrow \arg\max_{c \in \mathcal{C}} \text{Score}[c]$ ;    // Select best context strategy
15:    Generate revised plan $S$ using $c_t$;
16:    $\mathcal{P} \leftarrow \mathcal{P} \cup \{S\}$ ;    // Append revised plan to candidate pool
17:    Compute reward $\nu_t$ using Eq. 4;
18:    Update::;
19:     $R[c_t] \leftarrow R[c_t] + \nu_t$;
20:     $N[c_t] \leftarrow N[c_t] + 1$,   $N \leftarrow N + 1$;
21:     $\text{count}_k[c_t] \leftarrow \text{count}_k[c_t] + 1$;
22:   **end**
23:   $T \leftarrow \text{len}(S)$ ;     // Update plan length after success
24: **end**

---

### B.2 Task-Specific Reward

We define the task-specific reward $\nu_{\text{local}}$ as a weighted aggregation of five interpretable metrics, each designed to capture different aspects of reasoning quality at step $t$. These metrics jointly assess factual accuracy, semantic relevance, and reasoning efficiency, based on the alignment between the revised plan $S^{(t)}$, the question $Q$, and the knowledge graph feedback $O_t$.

- **Validation:** Measures the factual correctness of the predicted observation $Pred\_O_t$ using a contradiction classifier. Specifically, we employ the `DeBERTa-large-MNLI` model to assess whether the prediction contradicts or aligns with the KG feedback. Additional penalties are applied for degenerate outputs such as "none," "unknown," or empty spans.

- **Quality:** Evaluates the semantic relevance between the predicted observation $Pred\_O_t$ and the input question $Q$.

- **Question Alignment:** Measures the overall coherence of the reasoning step $S^{(t)}$ with respect to the original question $Q$. This is computed via embedding-level similarity to ensure that each revision remains question-centric.

- **Efficiency:** Penalizes redundancy across reasoning steps by comparing the semantic similarity of the current thought $s_t$ with all previous thoughts $s_{1:t-1}$. High overlap in meaning reduces the reward, encouraging non-redundant, progressive reasoning.

- **Thought Coherence:** Assesses whether the current reasoning step is adequately grounded in the KG feedback $O_t$. We calculate the similarity between the step's "Thought" component and the KG observations using the same embedding model.

For all similarity-based metrics, we compute cosine similarity between sentence embeddings obtained from a pretrained model (`msmarco-bert-base-dot-v5`).

### B.3 Implementation Details

#### B.3.1 Hyperparameter settings

We summarize below the hyperparameter settings used in our context selection and reward computation modules.

For the UCB-based strategy selection, we use an exploration coefficient of $\alpha = 1.4$ to balance the trade-off between exploration and exploitation. With regard to the three context-aware bonus terms, we adopt default values based on practical intuition rather than extensive search. Specifically, the entropy-based bonus is scaled by $\lambda_{\text{ent}} = 0.1$, encouraging exploration under high uncertainty. The KG-based redundancy penalty is set with $\lambda_{\text{KG}} = 0.1$, discouraging repetitive or overly deep retrieval. For the diversity bonus, we assign strategy-specific coefficients $\lambda_{\text{div}}^{(c)} = \{0.05, 0.1, 0.2\}$ for the Local, Lookahead, and Global strategies respectively, reflecting varying tolerance for repetition. To evaluate the impact of these weights, we further perturbed them by $\pm 20\%$ and observed only marginal changes in accuracy ($\leq 1.2\%$) and stable strategy distributions in Appendix E.5, confirming that VoG's performance is robust to hyperparameter variation.

The expected reasoning depth is set to $T_d = 5$ for CWQ, and $T_d = 3$ for both WebQSP and WebQuestions, reflecting the variation in question complexity and the structural depth of their corresponding knowledge graphs. For instance, CWQ typically requires longer and more compositional reasoning plans compared to WebQuestions. In reward aggregation, we use an entropy-based interpolation factor $\beta = 0.4$ for CWQ, and $\beta = 0.2$ for both WebQSP and WebQuestions, to control the weighting between task-specific and confidence-based reward components. The final plan selection threshold is set to 0.73 for CWQ and 0.77 for the other two datasets, tuning the sensitivity of plan acceptance to dataset-specific characteristics. Above task-specific values were chosen based on the average plan lengths observed from a small sample of development questions, and no further tuning was performed.

#### B.3.2 Experiment Settings

In our experiments, we evaluate VoG using three different language models: GPT-3.5 and GPT-4 accessed via the OpenAI API,[1] and Qwen2.5, which is deployed locally.[2] We set the temperature to 0.3 to reduce generation randomness, and restrict the maximum number of generation tokens to 1024 across all experiments for consistency. The experiments are conducted on an NVIDIA A800 GPU server.

## C Datasets

We evaluate our method on three widely-used knowledge graph question answering (KGQA) benchmarks: **ComplexWebQuestions** (Talmor and Berant, 2018), **WebQSP** (Yih et al., 2016), and **WebQuestions** (Berant et al., 2013). All datasets are constructed on the external knowledge graph from Freebase(Bollacker et al., 2008) and require multi-hop reasoning to reach the answer. The statistics of the datasets used in this paper are shown in Table 5.

Table 5: Dataset statistics.

| Dataset | Answer Type | Train | Test |
|---|---|---|---|
| ComplexWebQuestions | Entity | 27,734 | 3,531 |
| WebQSP | Entity / Number | 3,098 | 1,639 |
| WebQuestions | Entity/Number | 3738 | 2,032 |

## D Baseline Descriptions

We compare VoG against three categories of baselines:

---

[1] https://platform.openai.com/docs
[2] https://huggingface.co/Qwen

### D.1 LLM-ONLY BASELINES

These methods test the inherent reasoning capabilities of LLMs without external knowledge.

- **IO Prompting** (Brown et al., 2020): This approach performs few-shot prompting using direct input-output examples without any intermediate reasoning steps.
- **Chain-of-Thought (CoT)** (Trivedi et al., 2023): It generates intermediate reasoning chains that help the model arrive at a more accurate final answer.
- **Self-Consistency (SC)** (Wang et al., 2022): This method samples multiple CoT reasoning chains and selects the most consistent final answer through majority voting.

### D.2 FINE-TUNED METHODS

These methods incorporate KG information via supervised learning or fine-tuning strategies.

- **UniKGQA** (Jiang et al., 2022) unifies KG path retrieval and reasoning by introducing a pre-training objective based on question-relation matching, enabling shared representation learning.
- **DECAF** (Yu et al., 2023) linearizes the knowledge base into text-like sequences and retrieves relevant subgraphs using text-based retrieval. It jointly generates both logical forms and direct answers, combining the strengths of symbolic and generative reasoning.
- **KD-CoT** (Wang et al., 2023b) introduces a retrieval-augmented CoT framework, where an LLM queries a retriever for external knowledge and refines its reasoning chains based on returned answers, improving accuracy and credibility.
- **RoG** (Luo et al., 2024b) employs a fine-tuned LLM to generate reasoning plans based on the KG. These plans guide the retrieval of faithful evidence from the KG, improving the factual alignment of the reasoning process.
- **KG-Agent** (Zhao et al., 2024) uses a fine-tuned planner within a tool-augmented agent framework, enabling iterative interaction with KG APIs for multi-hop question answering.
- **GNN-RAG** (Mavromatis and Karypis, 2024) employs lightweight graph neural networks to score nodes and their neighborhoods based on question relevance, enabling effective retrieval over long-range KG contexts.
- **SubgraphRAG** (Li et al., 2025) employs a lightweight MLP with parallel triple-scoring and directional distance encoding to efficiently construct flexible subgraphs tailored to each query and model capacity.

### D.3 AGENT-BASED KG-AUGMENTED METHODS

These methods utilize LLMs as agents to guide reasoning over KGs through prompting, without requiring fine-tuning.

- **ToG** (SUN et al., 2024) treats an LLM as an agent that performs beam search over the KG. It iteratively expands and scores candidate paths to discover the most promising reasoning trajectories.
- **PoG** (Chen et al., 2024) decomposes the input question into structured subgoals, which are then used to guide step-by-step retrieval and reasoning. Additional memory and reflection mechanisms are introduced to enhance coherence and accuracy.

We emphasize that VoG is a model-agnostic framework that does not require fine-tuning, and is directly compatible with both open-source and proprietary LLMs.

## E ADDITIONAL ANALYSIS AND ABLATION STUDY

In this section, we provide a deeper ablation to further examine the behavior and effectiveness of the proposed VoG framework beyond the main experimental results. All experiments are conducted with GPT-3.5 for consistency. Together, these ablation studies offer fine-grained insights into how each design choice contributes to VoG's overall accuracy, robustness, and efficiency.

### E.1 REVISION SIGNAL ANALYSIS

To better understand the distribution and necessity of revision across reasoning steps, we analyze how often the verification module flags reasoning steps for revision. We compare the revise signals generated by the LLM-based verifier (GPT-3.5) and a lightweight PLM-based verifier (e.g., DeBERTa-based NLI model(He et al., 2021)). Figure 5 visualizes the distribution of revision signals across three datasets, segmented by the source of verification. The outer ring distinguishes between samples that triggered a revision signal and those that did not, while the inner ring further breaks down the revision-triggering cases into those suggested exclusively by the LLM verifier, the PLM verifier, or by both. This visualization highlights the complementary nature of the two verifiers, as well as the relative proportion of agreed versus disagreed signals.

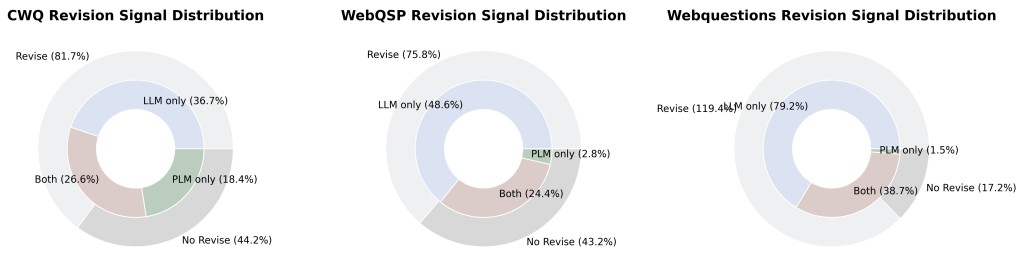

Figure 5: Distribution of revision signals across datasets. The outer ring shows the overall proportion of examples requiring revision, while the inner ring indicates the source of the revision signal (LLM only, PLM only, or both).

### E.2 DETAILED EFFICIENCY ANALYSIS

To provide a finer-grained view of efficiency, we further analyze the token usage distribution across different stages of VoG. Specifically, we separate the tokens consumed by planning, retrieval, verification, and revision, and report their proportions relative to the total tokens. Figure 6 presents the results on CWQ and WebQSP with GPT-3.5, based on a representative sample of test data. We find that initial planning consistently accounts for only a small fraction of the total tokens, whereas the relative costs of retrieval, verification, and revision vary across datasets.

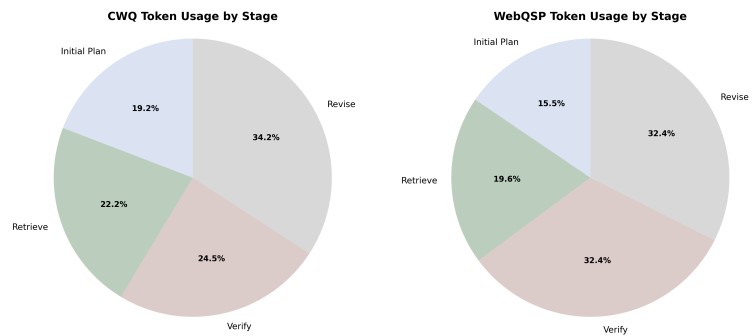

Figure 6: Stage-wise token distribution of VoG on CWQ and WebQSP. Each slice shows the proportion of tokens consumed by initial planning, retrieval, verification, and revision relative to the total.

### E.3 STRATEGY-LEVEL PERFORMANCE ANALYSIS

In contrast to the isolated strategy end-to-end comparison in Section 4.3, here we analyze the effectiveness of different strategies under the adaptive MAB framework. We first compute the per-iteration *revision success rate*, defined as the percentage of revisions that successfully correct an

initial error, considering only the instances where the prior plan's answer was incorrect. As shown in Table 6, all three strategies show comparable revision success rates.

Table 6: Revision success rates (%) for each strategy, computed as the proportion of revisions that successfully corrected an error in each attempt.

| Strategy | CWQ | WebQSP | WebQuestions |
|---|---|---|---|
| Local | 26.0 | 38.6 | 27.0 |
| Lookahead | 25.9 | 36.8 | 27.9 |
| Global | 26.1 | 33.6 | 28.5 |

Beyond evaluating whether a revision leads to a correct final answer, we additionally assess whether the model can correct incorrect intermediate reasoning steps by leveraging ground-truth SPARQL execution paths. In this setup, we use GPT-3.5 to label each revised step as: (i) **CORRECT**, fully matching the gold reasoning path; (ii) **IMPROVED**, an improvement over an originally incorrect step; or (iii) **INCORRECT**. Table 7 summarizes the proportions of CORRECT and CORRECT+IMPROVED steps for CWQ and WebQSP, indicating that the revision module successfully revises a large majority of wrong intermediate steps.

Table 7: Step-level correctness of revised reasoning steps using ground-truth SPARQL paths.

| Metric | CWQ | WebQSP |
|---|---|---|
| CORRECT + IMPROVED | 85.21% | 99.50% |
| CORRECT only | 44.36% | 83.25% |

We then examine which strategy contributes to the correct answers produced under the MAB setting. Figure 7 reports the proportion of correctly answered questions attributed to each strategy, highlighting their complementary roles in the overall framework.

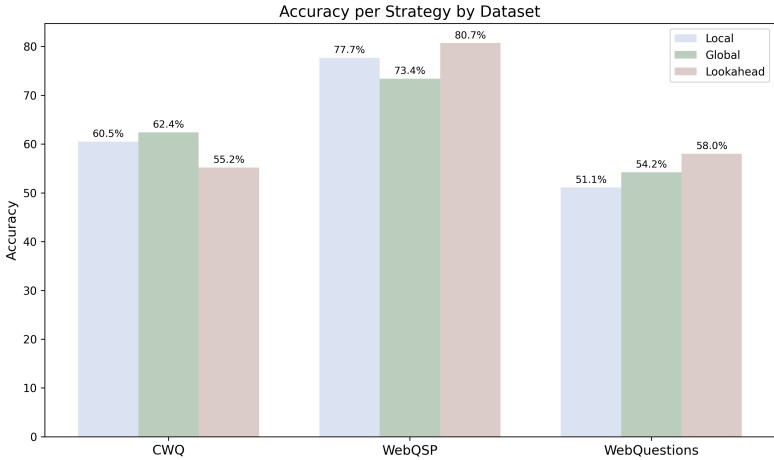

Figure 7: Strategy-specific answer accuracy under the MAB framework across three datasets. Each bar represents the proportion of correctly answered questions attributed to a given strategy.

### E.4 ABLATION ON BONUS TERMS

We evaluate the contribution of each component in our KG-aware UCB scoring mechanism by conducting ablation studies on the CWQ dataset. Specifically, we consider three variants: (i) removing all KG-aware prior terms, (ii) removing only the diversity-aware bonus, and (iii) removing both, effectively reducing our scoring function to the standard UCB formulation (Auer et al., 2002). Table 8 reports the performance degradation under each variant. The results demonstrate that incorporating

KG-specific priors and diversity control leads to more effective context selection, ultimately enhancing reasoning accuracy.

Table 8: Ablation study of UCB bonus terms across datasets.

| UCB Variant | CWQ | WebQSP | WebQuestions |
|---|---|---|---|
| Modified UCB (all bonuses) | **64.7** | **83.2** | **63.0** |
| w/o KG-aware priors | 64.2 | 77.7 | 60.2 |
| w/o Diversity | 63.8 | 77.3 | 59.4 |
| w/o both(UCB) | 63.0 | 74.7 | 58.9 |

## E.5 ROBUSTNESS OF MODIFIED UCB

To assess the robustness of our UCB formulation, we performed perturbation analysis on both the context-aware bonus terms and the exploration weight. Specifically, we perturbed the weights of these terms by $\pm 20\%$. Across all perturbation settings, VoG consistently outperforms other GPT-3.5-based baselines, with performance fluctuations remaining marginal ($\leq 1.2\%$ in accuracy). These results confirm that our approach is robust to moderate changes in weighting. Tables 9 and 10 report the accuracy, average plan length, and average strategy counts under each perturbation for CWQ and WebQSP, respectively.

Table 9: Robustness analysis of context-aware bonus weights on CWQ and WebQSP.

| Dataset | Weight Perturbation | Accuracy (%) | Avg Plan Length | Avg Strategy Counts (Local / Lookahead / Global) |
|---|---|---|---|---|
| **CWQ** | +20% | 63.5 | 3.49 | 1.60 / 1.22 / 1.21 |
| | −20% | 63.6 | 3.52 | 1.59 / 1.21 / 1.15 |
| | 0% | 64.7 | 3.62 | 1.69 / 1.26 / 1.22 |
| **WebQSP** | +20% | 82.4 | 2.76 | 1.35 / 0.83 / 0.70 |
| | −20% | 83.2 | 2.78 | 1.34 / 0.81 / 0.66 |
| | 0% | 83.2 | 2.80 | 1.35 / 0.80 / 0.63 |

Table 10: Robustness analysis of exploration weights on CWQ and WebQSP.

| Dataset | Weight Perturbation | Accuracy (%) | Avg Plan Length | Avg Strategy Counts (Local / Lookahead / Global) |
|---|---|---|---|---|
| **CWQ** | +20% | 64.3 | 3.53 | 1.65 / 1.32 / 1.16 |
| | −20% | 63.6 | 3.62 | 1.61 / 1.29 / 1.15 |
| | 0% | 64.7 | 3.62 | 1.69 / 1.26 / 1.22 |
| **WebQSP** | +20% | 82.4 | 2.73 | 1.36 / 0.86 / 0.79 |
| | −20% | 83.3 | 2.79 | 1.33 / 0.84 / 0.61 |
| | 0% | 83.2 | 2.80 | 1.35 / 0.80 / 0.63 |

## E.6 ABLATION STUDY ON REWARD DESIGN

We conduct two sets of ablation experiments to evaluate the contribution of different reward components used in our MAB-based context selection.

**Task-Specific vs. Confidence-Based Reward.** We first compare the performance of the task-specific reward and confidence-based reward in terms of their ability to guide the selection of high-quality plans. Specifically, we report the accuracy of the highest-scoring plan from the full set of candidates $\mathcal{P}$ under each reward formulation. Note that this includes all plans proposed throughout the iterative reasoning process, regardless of whether they were ultimately accepted or discarded. As shown in Table 11, the confidence-based reward generally achieves better performance, indicating a stronger alignment with answer correctness. Additionally, we compute the mean entropy of answer

distributions aggregated over all revision steps. This metric captures the average level of uncertainty during reasoning across datasets. It is important to distinguish this analysis from the actual decision process in our MAB-based controller. During inference, the final answer is produced by the plan selected at the last revision step. Specifically, the one that exceeds the reward threshold and is chosen based on dynamic strategy selection. In contrast, this ablation purely evaluates each reward's ability to assign higher scores to more accurate plans within the full candidate pool.

Table 11: Accuracy (%) of the plan selected by different reward methods, and corresponding answer entropy.

| Reward Method | CWQ | WebQSP | WebQuestions |
|---|---|---|---|
| Task-specific reward | 56.5 | 75.9 | 59.1 |
| Confidence reward | 60.4 | 79.1 | 62.4 |
| Entropy (avg.) | 0.37 | 0.34 | 0.32 |

**Component-wise Ablation of Task-Specific Reward.** To understand the role of each component in the task-specific reward design, we perform a leave-one-out ablation study. As shown in Table 12, removing any individual component leads to a drop in accuracy, confirming the necessity of all five elements. These results highlight the importance of penalizing hallucinated or contradictory outputs and aligning the reasoning step with the input query.

Table 12: Ablation Accuracy (%) on Task-Specific Reward across Datasets

| Setting | CWQ | WebQSP | WebQuestions |
|---|---|---|---|
| Full Reward | **56.50** | **50.61** | **75.60** |
| w/o Quality | 52.41 | 49.47 | 71.45 |
| w/o Thought Completion | 52.66 | 49.82 | 70.35 |
| w/o Efficiency | 52.49 | 50.53 | 71.29 |
| w/o Question Alignment | 52.15 | 49.47 | 70.98 |
| w/o Validation | 51.13 | 48.07 | 68.61 |

### E.7 TRADE-OFF ANALYSIS BETWEEN COMPLEXITY AND PERFORMANCE

This subsection provides a quantitative study of the trade-offs between model complexity and performance in the VoG framework. We examine several simplified variants of VoG to understand how individual components contribute to accuracy and computational cost. Specifically, we explore the effect of replacing the UCB with fixed local, global, and lookahead revision strategies, denoted as *local revision*, *global revision*, and *lookahead revision*, respectively, as well as substituting the *verify agent* with a *PLM-only verifier*. Table 13 reports the resulting accuracy changes, token usage differences, and the corresponding maximum LLM call complexity on the CWQ dataset.

Table 13: Trade-off analysis of simplifications in VoG on CWQ dataset.

| Simplification | $\Delta$Acc | $\Delta$Token Usage | Maximum LLM Call (Simplified / Full) |
|---|---|---|---|
| *local revision* | $-4.6\%$ | $-25.0\%$ | $(1 + D(2k_{\max} + 2))/(1 + D(2k_{\max} + 1 + T))$ |
| *global revision* | $-4.5\%$ | $-29.1\%$ | $(1 + D(2k_{\max} + 2))/(1 + D(2k_{\max} + 1 + T))$ |
| *lookahead revision* | $-1.1\%$ | $-21.1\%$ | $(1 + D(2k_{\max} + 2))/(1 + D(2k_{\max} + 1 + T))$ |
| *PLM-only verifier* | $-2.2\%$ | $-21.1\%$ | $(1 + D(2k_{\max} + T))/(1 + D(2k_{\max} + 1 + T))$ |

**Note:** $D$ is the depth of reasoning plan, $k_{\max}$ is the maximum width given in Algorithm 1, and $T$ is the revision times.

# F ADDITIONAL CASE STUDY

## F.1 ADDITIONAL CASES

Beyond the representative case analyzed in Section 4.5, we also examine additional examples that highlight how different revision strategies become effective under different conditions. Figure 8 presents two additional cases using Qwen2.5-7B. The upper case shows how future relations provide useful guidance to revise the plan, while the lower case demonstrates that they may instead introduce distraction. We also observe that the *Local* strategy often works well in earlier steps when explicit evidence is available and competing candidates exist, as in the lower case, where lightweight local correction prevents unnecessary detours.

Together, these examples highlight both the potential and risks of relying on broader context, further underscoring the importance of adaptive strategy selection. These examples further highlight the difficulty of predicting which context will be most effective, reinforcing the importance of dynamic strategy selection. We also note that although intermediate answers at non-terminal steps are not final outputs, their errors can mislead subsequent reasoning and incur unnecessary computational cost.

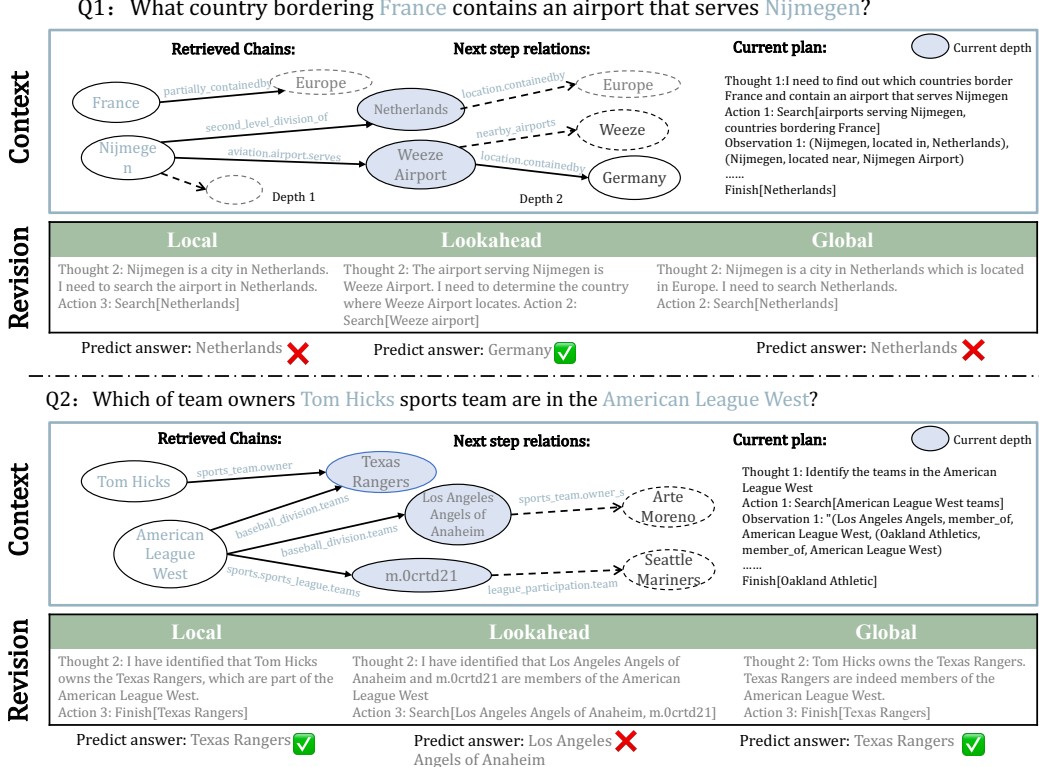

Figure 8: Additional two case studies with Qwen2.5-7B, illustrating how different revision strategies become effective under different conditions.

## F.2 CASE COMPARISON WITH BASELINES

While Section 4.5 presents a strategy-level case study highlighting VoG's internal context selector, here we provide a complementary example comparing VoG against two external agent-based baselines, ToG and PoG. Figure 9 illustrates the question, retrieved triplets, and intermediate reasoning process for each method. This example highlights how VoG's iterative verification and context-aware revision mechanisms help mitigate hallucinations and enable correction of intermediate errors.

Both ToG and PoG fail to produce the correct answer in this example, but for different reasons. **ToG**, which conducts beam search over KG triplets by prompting an LLM to score candidate paths, retrieves facts such as *(France, location.location.containedby, Europe)* and *(Nijmegen, second_level_division,*

*Netherlands).* However, lacking structured planning or subgoal decomposition, it prunes valid paths prematurely and incorrectly concludes that insufficient information is available.

**PoG** performs better in retrieval, identifying facts like *(Nijmegen, location.location.nearby_airports, Weeze Airport).* However, it suffers from LLM hallucination, prematurely terminating the reasoning process and erroneously predicting the Netherlands as the final answer without validating supporting facts.

In contrast, **VoG** successfully answers the question by employing three key mechanisms: (i) an initial reasoning plan that explicitly outlines the intended depth or retrieval, (ii) stepwise verification that checks the factual correctness of each intermediate step against retrieved KG triplets, and (iii) a context-aware revision strategy, such as *lookahead*, which dynamically adapts retrieval and plan updates. Moreover, because VoG performs verification and revision along the entire reasoning chain, it is more tolerant to locally missing evidence *"Unnamed Entity"* as illustrated in ToG's case than baselines that rely solely on local decisions. These allow VoG to refine incorrect steps and extend reasoning depth when necessary, leading to a factually grounded and correct answer.

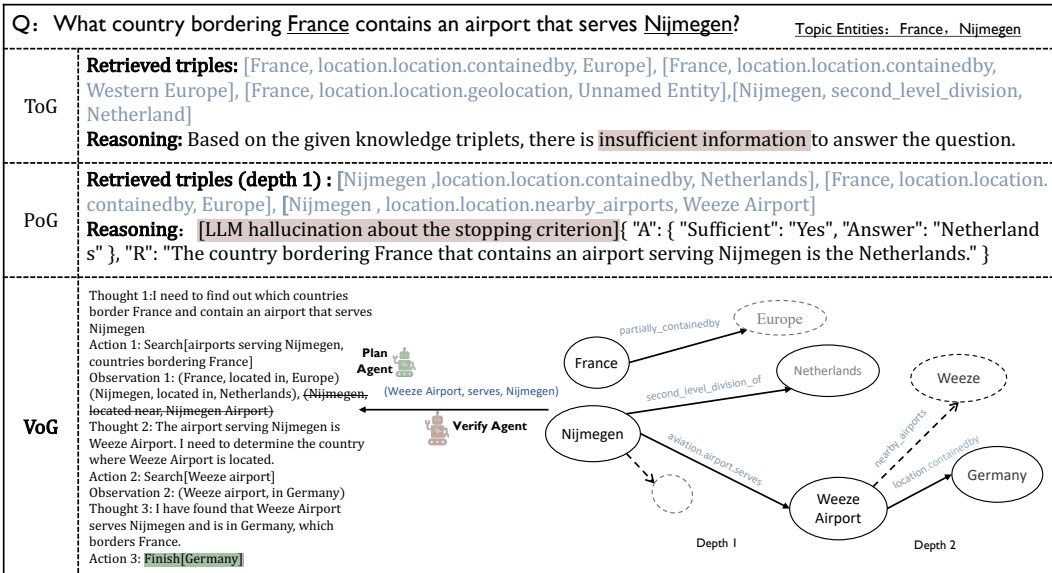

Figure 9: Case study comparing VoG, ToG, and PoG.

## F.3 ANALYSIS OF RECOVERED FAILURE TYPES

To further investigate which types of reasoning failures benefit most from VoG's verification mechanism, we conducted an additional error analysis. Failure cases were categorized into four types via annotation assisted by GPT-3.5, with spot-checking for consistency. Table 14 summarizes the results.

Table 14: Distribution of recovered failure cases using GPT-3.5 annotation.

| Failure Type | Description | Proportion (%) |
|---|---|---|
| Hallucinated fact | LLM fabricates a fact not supported by KG evidence. | 37.5 |
| Incomplete evidence | Answer returned before all necessary evidence is collected and verified. | 27.9 |
| Entity disambiguation error | Incorrect entity chosen when multiple candidates exist. | 21.7 |
| Query intent drift | Reasoning plan deviates from the original query intent. | 12.9 |

We find that hallucinated facts and incomplete evidence are the most effectively recovered failure types, which aligns with our main claim that stepwise verification and context-aware revision help mitigate hallucination and premature stopping.

# G  CROSS-KG GENERALIZATION

To examine VoG's transferability beyond Freebase, we conduct two generalization experiments on both an open-domain KG with a different schema and a domain-specialized biomedical KG.

**Generalization to Wikidata.**  We evaluate VoG on Wikidata (Vrandečić and Krötzsch, 2014) to assess cross-KG robustness. We directly replace the KG-specific retrieval interface while keeping the reasoning and verification modules unchanged. Experiments are conducted on the QALD-10 (Santana et al., 2022) benchmark, showing that VoG achieves competitive performance without any task-specific training as shown in Table 15.

Table 15: The results comparison of different methods on the QALD10-en dataset.

| Model | Method | Accuracy (%) |
|---|---|---|
| Fine-tuned | SPARQL-QA (Santana et al., 2022) | 45.4 |
| GPT 3.5 | ToG | 50.2 |
| GPT 3.5 | VoG | **57.4** |

**Generalization to Biomedical Knowledge.**  Generalizing across domain KGs remains an open challenge in the field due to the scarcity of high-quality multi-hop QA datasets outside open-domain settings. Existing biomedical datasets are typically yes/no or long free-form question answering, making controlled evaluation difficult. To enable a preliminary assessment, we construct a small multi-hop biomedical QA benchmark based on Hetionet (Himmelstein et al., 2017), a UMLS-derived KG containing 47,031 nodes and 2,250,197 edges. We extract multi-hop relational paths, filter them for semantic coherence, and use GPT-3.5 rewriting to obtain 200 naturally phrased biomedical questions (e.g., "Which genes interact with genes that are downregulated by Dacarbazine?"). Owing to the reduced contextual variability in biomedical queries, we adopt our local revision strategy for this study. Table 16 shows the accuracy gain of VoG on our test set.

Table 16: The results on our test set.

| Setting | Accuracy (%) |
|---|---|
| GPT 3.5 | 1.0 |
| VoG | 19.5 |
| **Gain** | **+18.5** |

We further evaluate VoG on two biomedical datasets, PubMedQA (Jin et al., 2019) and BioASQ (Krithara et al., 2023), both consisting of yes/no type questions that do not require multi-hop reasoning. The experiment is conducted under the sparse-KG setting introduced in GIVE (He et al., 2025), using a UMLS-derived KG with only 135 nodes and 5,877 edges, which severely limits the available triplets and path candidates. Table 17 shows the results comparison in terms of accuracy, average token usage, and average inference time.

Table 17: The results comparison of GIVE, ToG, and VoG on PubMedQA and BioASQ.

| Dataset | Method | Accuracy (%) | Avg Tokens | Avg Time (s) |
|---|---|---|---|---|
| | GIVE | 53.6 | 14,701.5 | 33.6 |
| PubMedQA (Jin et al., 2019) | ToG | 17.6 | 12,805.3 | 15.9 |
| | VoG | 45.3 | **1,653.2** | **12.1** |
| | GIVE | 88.2 | 8,050.3 | 15.3 |
| BioASQ (Krithara et al., 2023) | ToG | 18.0 | 7,070.2 | 10.3 |
| | VoG | **93.5** | **1,862.6** | 14.1 |

These results demonstrate that VoG can generalize beyond Freebase to KGs with different structural and semantic characteristics. This transferability is primarily enabled by VoG's training-free design,

where retrieval relies on pretrained text embeddings that carry semantically transferable signals across heterogeneous schemas. The inherent cross-domain generalization capability of LLMs also stabilizes VoG's reasoning on specific domains even when domain knowledge is sparse or noisy.

## H  LIMITATION

Despite the strong performance of VoG in multi-hop KG reasoning, several limitations remain:

**Reliance on KG Completeness:**   VoG assumes access to a reliable and sufficiently complete KG. However, real-world KGs that constructed from web corpora are often noisy or incomplete, which may lead to retrieval failures or factual errors. In future work, we plan to incorporate KG confidence scores or external sources to mitigate such issues.

**Frozen LLM-based Verifier:**   In this work, we leverage frozen LLMs and PLMs as verifiers, which may be limited by their pre-training distributions and lack of task-specific tuning. As a result, subtle inconsistencies may go undetected during stepwise verification. In the future, we plan to explore fine-tuning LLMs to act as more reliable verification modules and enable stronger factual validation and more accurate revision signals.

**Basic Context Selection Granularity:**   Our current context selector relies on signals like entropy and step. Incorporating structural signals from the KG, such as node centrality or subgraph coherence, may offer finer-grained control over reasoning revision and is worth exploring in future.

## I  THE USE OF LARGE LANGUAGE MODELS

Large language models (LLMs) were used in this work as supportive tools to polish the writing. They assisted with grammar, clarity, and style, but did not contribute to the design of the methodology, implementation of experiments, or interpretation of results. Their role was limited to improving readability, without generating original research contributions.

