# OpenReview forum: "VoG: Enhancing LLM Reasoning through Stepwise Verification on Knowledge Graphs"
_ICLR.cc/2026/Conference — ICLR 2026 Poster_

### Official Review · Reviewer_YG4Q · 2025-10-27

**Soundness:** 3
**Presentation:** 3
**Contribution:** 3
**Rating:** 6
**Confidence:** 4

**Summary:**

The paper proposes Verify-on-Graph (VoG), a framework that enhances Large Language Models' reasoning by integrating knowledge graphs for reliable inference. VoG features iterative retrieval, verification, and adaptive planning refinement to boost accuracy and efficiency. Experiments on three benchmark datasets demonstrate its effectiveness in improving reasoning performance.

**Strengths:**

1. VoG introduces a novel framework that enables stepwise verification and planning refinement on KGs to mitigate error propagation during multi-hop reasoning.
2. VoG proposes a KG-aware multi-armed bandit (MAB) mechanism for adaptive context selection, which is a valuable contribution for dynamically determining context information for refining reasoning plans.
3. Experiment on three benchmark datasets shows that VoG significantly outperforms existing methods, demonstrating its effectiveness in enhancing the reasoning capabilities of LLMs.

**Weaknesses:**

1. The architecture of VoG seems complex with multiple components. While each component is motivated and performance improvements are shown, the overall complexity may hinder practical adoption. The authors should provide more analysis on the trade-offs between complexity and performance gains, and possibly explore simplifications.
2. UCB hyperparameter sensitivity is not deeply analyzed — how robust is the method to different scaling of entropy or exploration bonuses?
3. No evaluation beyond Freebase / KGQA — unclear whether VoG generalizes to non-Freebase or domain-specialized KGs (e.g., ConceptNet , biomedical UMLS).
4. The proposed method relies on retrieving evidence from the KG, but it is unclear how VoG handles cases where valid evidence/fact are missing from the KG.

**Questions:**

1. How does VoG handle valid but missing KG facts during reasoning?
2. How sensitive is the performance to the UCB hyperparameters?
3. Could VoG generalize beyond Freebase, e.g., ConceptNet or domain-specific KGs?

---

> ### Author Response · Authors · 2025-11-21
>
> We appreciate the reviewer's insightful feedback. Below, we address the reviewer’s comments in detail.
>
> > **W1:** The authors should provide more analysis on the trade-offs between complexity and performance gains, and possibly explore simplifications.
>
> **A1:** We appreciate the constructive concern regarding the trade-offs between complexity and performance gains of the VoG framework. **The trade-off analysis following your suggestion is conducted on the CWQ dataset and shown as follows.**
>
> | Simplification                              | Description                                                  | Performance Drop (ΔAcc) | ΔToken Usage | Maximal LLM Call (Simplified / Full)                    |
> | ------------------------------------------- | ------------------------------------------------------------ | ----------------------- | ------------ | ------------------------------------------------------- |
> | Simplification 1.1: Local-Revision-Only     | Replace UCB with fixed Local context                         | −4.6%                   | -25.0%       | $(1 + D·(2k_{max}  + 2))  / (1 + D·(2k_{max} + 1 + T))$ |
> | Simplification 1.2: Global-Revision-Only    | Replace UCB with fixed Global context                        | −4.5%                   | -29.1%       | $(1 + D·(2k_{max}  + 2))  / (1 + D·(2k_{max} + 1 + T))$ |
> | Simplification 1.3: Lookahead-Revision-Only | Replace UCB with fixed Lookahead context                     | −1.1%                   | -21.1%       | $(1 + D·(2k_{max}  + 2))  / (1 + D·(2k_{max} + 1 + T))$ |
> | Simplification 2: PLM-only Verification     | Replace LLM-based Verify Agent with PLM-based verifier only (see Appendix E.1) | −2.2%                   | -21.1%       | $(1+D·(2k_{max} + T)) / (1 + D·(2k_{max} + 1 + T))$     |
>
> *Note: D refers to length of depth while T refers to the revision times.  _k_max_ is given in Appendix A.2*
>
> In practice, we also explore adopting Simplification 1.1 in our generalization experiment provided below. We have incorporated this analysis into our revised manuscript (see Appendix E.7).
>
> > **W2&Q2:** UCB hyperparameter sensitivity is not deeply analyzed. how robust is the method to different scaling of **entropy** or **exploration** bonuses?
>
> **A2:** Thanks for raising this point. The experiment following your advice is added to demonstrate the robustness of our method under different scaling of entropy or exploration bonuses, as shown below:
>
> - Sensitivity analysis on exploration bonus:
>
> | **Weight Perturbation** | **Dataset** | **Accuracy (%)** | **Avg Plan Length** | **Avg Strategy Counts (Local / Lookahead / Global)** |
> | ----------------------- | ----------- | ---------------- | ------------------- | ---------------------------------------------------- |
> | +20%                    | CWQ         | 64.3             | 3.53                | 1.65 / 1.32 / 1.16                                   |
> | −20%                    | CWQ         | 63.6             | 3.62                | 1.61 / 1.29 / 1.15                                   |
> | 0%                      | CWQ         | 64.7             | 3.62                | 1.69 / 1.26 / 1.22                                   |
> | +20%                    | WebQSP      | 82.4             | 2.73                | 1.36 / 0.86 / 0.79                                   |
> | −20%                    | WebQSP      | 83.3             | 2.79                | 1.33 / 0.84 / 0.61                                   |
> | 0%                      | WebQSP      | 83.2             | 2.80                | 1.35 / 0.80 / 0.63                                   |
>
> - Sensitivity analysis on entropy bonus:
>
> | **Weight Perturbation** | **Dataset** | **Accuracy (%)** | **Avg Plan Length** | **Avg Strategy Counts (Local / Lookahead / Global)** |
> | ----------------------- | ----------- | ---------------- | ------------------- | ---------------------------------------------------- |
> | +20%                    | CWQ         | 64.1             | 3.61                | 1.65 / 1.33 / 1.24                                   |
> | −20%                    | CWQ         | 63.7             | 3.48                | 1.60 / 1.22 / 1.19                                   |
> | 0%                      | CWQ         | 64.7             | 3.62                | 1.69 / 1.26 / 1.22                                   |
> | +20%                    | WebQSP      | 82.8             | 2.78                | 1.37 / 0.83 / 0.73                                   |
> | −20%                    | WebQSP      | 83.4             | 2.71                | 1.36 / 0.95 / 0.72                                   |
> | 0%                      | WebQSP      | 83.2             | 2.80                | 1.35 / 0.80 / 0.63                                   |
>
> We reported the other bonus terms in our original submission, and the results presented above have been integrated into our robustness study, helping to provide a more comprehensive analysis (see Appendix E.5).

---

> > ### Author Response · Authors · 2025-11-21
> >
> > > **W3&Q3:** No evaluation beyond Freebase / KGQA. Could VoG generalizes to **non-Freebase** or **domain-specialized KGs** (e.g., ConceptNet , biomedical UMLS)?
> >
> > **A3:** Thanks for this insightful concern on VoG's generalization to non-Freebase or domain-specialized KGs such as ConceptNet or UMLS. We address this by **conducting two additional experiments to evaluate VoG on both a non-Freebase general-domain KG and a UMLS-derived biomedical KG**.
> >
> > - Generalization to a **non-Freebase KG**:
> >
> >   To examine VoG’s effectiveness on a non-Freebase KG, we adapted VoG to Wikidata and evaluated it on the QALD-10 [1] benchmark. As Wikidata remains an open-domain KG, we only replaced the KG-specific retrieval interface without requiring prompt redesign. VoG achieves strong performance compared to previous training-based and training-free methods:
> >
> >   | **Model**   | **Method**    | **Accuracy(%)** |
> >   | ----------- | ------------- | --------------- |
> >   | Fine-Tuned  | SPARQL-QA [1] | 45.4            |
> >   | ChatGPT 3.5 | ToG           | 50.2            |
> >   | ChatGPT 3.5 | VoG           | 57.4            |
> >
> > - Generalization to a **biomedical KG**:
> >
> >   Currently, generalizing to domain-specialized KGs remains an open challenge in the field, partly because most domain-specific KGs (e.g., biomedical or scientific KGs) lack high-quality multi-hop QA datasets and existing resources are often yes/no style or have long free-form answers. Here we try our best to build a small biomedical multi-hop QA test set using Hetionet [2], a UMLS-derived KG with 47,031 nodes and 2,250,197 edges. We extract multi-hop relational paths, filter them for coherence, and use LLMs to rewrite them into 200 naturally phrased biomedical questions (e.g., “Which genes interact with genes that are downregulated by Dacarbazine?”). Because biomedical questions exhibit much simpler contextual structure than open-domain ones, we use a simplified adaptation and apply VoG’s local revision strategy. The results are as follows:
> >
> >   | **Setting**                   | **Accuracy (%)** |
> >   | ----------------------------- | ---------------- |
> >   | Initial GPT Answer (baseline) | 1.0              |
> >   | VoG Final Answer (ours)       | 19.5             |
> >   | Gain                          | +18.5            |
> >
> > VoG’s cross-KG transferability primarily comes from two design principles:
> >
> > - **Training-free pipeline**: All components in VoG are training-free. Retrieval relies on pretrained text embeddings, which capture transferable semantics across heterogeneous KGs, enabling effective pruning even without domain-specific tuning.
> > - **LLM-driven planning + verify–revise loop**: LLMs possess inherent **cross-domain reasoning ability**, which our planning module leverages. The verify–revise process then stabilizes the reasoning trajectory with retrieved KG evidence.
> >
> > We have incorporated these additional results and discussion into the revised manuscript (see Appendix G). In future work, we plan to develop higher-quality multi-hop datasets across diverse KGs and conduct more systematic evaluations to further strengthen this line of exploration.

---

> > > ### Author Response · Authors · 2025-11-21
> > >
> > > > **W4&Q1:** The proposed method relies on retrieving evidence from the KG, but it is unclear how VoG handles cases where valid evidence/facts are missing from the KG.
> > >
> > > **A4:** We appreciate this valuable concern about VoG's behaviors when valid evidence or facts are missing from the KG. Specifically, **there are two types of cases that VoG handles differently**.
> > >
> > > - **Partially Missing:** In our main benchmarks (e.g., CWQ, WebQSP), where the underlying Freebase KG is sufficiently large, the incompleteness we encounter is that facts are locally missing while a valid multi-hop path still exists in the KG. For example, in Fig. 1(2), the intermediate entity "m.0h_1h3x" lacks a human-readable label in the KG, but the underlying path still exists, such as "Noble patria, tu hermosa bandera → government.national_anthem_of → m.0h_1h3x → government.national_anthem_of_a_country.country → Costa Rica".
> > >
> > >   In such cases, rather than directly outputting an answer or stopping retrieval as in prior works, VoG is required to execute the entire planned reasoning chain. Even if an intermediate step cannot be verified or incorrectly revised due to missing facts, later steps can still be verified and revised using the subsequent KG facts together with the evolving reasoning context. Because the planning stage provides a globally coherent trajectory along which VoG performs stepwise verification, the framework is more tolerant to locally missing evidence than methods that make purely local decisions or terminate when a single hop fails (see lines 86-87 of our updated version).
> > >
> > > - **Entirely Missing:** When generalizing to smaller or sparser KGs, there arose cases where the entire valid path is missing from the KG. We explicitly tested VoG under such sparse-KG conditions on a UMLS-derived KG with 135 nodes and 5,877 edges [3] (see Appendix G). In this regime, the planning module serves as a fallback when KG evidence is insufficient. When no valid evidence in KG contradicts VoG’s plan, VoG refrains from unnecessary revisions based on irrelevant facts and relies on the LLM's inherent cross-domain generlization capabilities. Retrieving on highly incomplete KGs remains an open challenge for KG-augmented LLM agents, and we plan to develop more suitable approaches for handling extremely sparse KGs in future work.
> > >
> > > We have incorporated the discussion into our revised manuscript (see Appendix F.2 and Appendix G).
> > >
> > > [1] Borroto, Manuel, et al. "SPARQL-QA enters the QALD challenge." *Proceedings of the 7th Natural Language Interfaces for the Web of Data (NLIWoD) co-located with the 19th European Semantic Web Conference, Hersonissos, Greece*. Vol. 3196. 2022.
> > > [2] Himmelstein, Daniel Scott, et al. "Systematic integration of biomedical knowledge prioritizes drugs for repurposing." *elife* 6 (2017): e26726.
> > > [3] He, Jiashu, et al. "Give: Structured reasoning of large language models with knowledge graph inspired veracity extrapolation." *arXiv preprint arXiv:2410.08475* (2024).

---

> > > > ### Comment · Reviewer_YG4Q · 2025-11-23
> > > >
> > > > Thank you to the authors for the detailed response, which addresses most of my concerns. I will maintain my positive score.

---

> > > > > ### Author Response · Authors · 2025-11-23
> > > > >
> > > > > We are glad that our revisions have addressed your concerns. We sincerely appreciate your thoughtful feedback and your effort in reviewing our work.

---

### Official Review · Reviewer_rU7C · 2025-10-31

**Soundness:** 4
**Presentation:** 4
**Contribution:** 4
**Rating:** 8
**Confidence:** 5

**Summary:**

This paper proposes VoG (Verification-over-Generation), a reasoning framework that enhances large language models (LLMs) with structured, knowledge-grounded verification and revision. Unlike previous reasoning or retrieval-augmented methods that rely solely on planning or evidence retrieval, VoG organizes reasoning into a Plan–Retrieve–Verify–Revise loop.

**Strengths:**

The paper introduces a unified Plan–Retrieve–Verify–Revise reasoning framework, emphasizing step-wise knowledge-grounded verification. This design effectively connects planning with factual correction and hallucination suppression, offering a more systematic approach than prior single-stage (planning-only or retrieval-only) methods. Compared to ToG/PoG agents, VoG achieves lower average token consumption and fewer reasoning turns while maintaining higher accuracy, supporting the paper’s claim of being lightweight and efficient.

**Weaknesses:**

The author’s efficiency analysis section is very convincing; however, I am still curious about the end-to-end latency, as its framework introduces more steps compared to ToG and PoG.

**Questions:**

See weakness above.

---

> ### Author Response · Authors · 2025-11-21
>
> We appreciate reviewer's valuable feedback. We are encouraged that the reviewer recognized the effectiveness of VoG's framework. We address the reviewer’s questions as follows:
>
> > **W1:** The author’s efficiency analysis section is very convincing; however, I am still curious about the end-to-end latency, as its framework introduces more steps compared to ToG and PoG.
>
> **A1:** We thank the reviewer for this comprehensive comment. The direct comparison on inference time across agent-based methods is added below:
>
> | **Dataset** | **Method** | **Time (s)** |
> | ----------- | ---------- | ------------ |
> | CWQ         | ToG        | 96.5         |
> |             | PoG        | 34.3         |
> |             | Ours       | 38.9         |
> | WebQSP      | ToG        | 63.1         |
> |             | PoG        | 16.8         |
> |             | Ours       | 17.9         |
>
> We further report the runtime breakdown across stages on sampled CWQ data as below:
>
> | **Plan** | **Retrieve** | **Verify** | **Revise** |
> | -------- | ------------ | ---------- | ---------- |
> | 6.1%     | 49.0%        | 6.9%       | 38.0%      |
>
> The retrieve stage accounts for the largest portion of runtime. Although VoG introduces additional steps compared to ToG, our design actually brings efficiency gains in retrieval. ToG performs beam-style exploration with a predefined breadth, which leads to the expansion of many irrelevant candidate paths. In contrast, VoG’s retrieval stage provides adaptive semantic filtering (see Appendix A.2), allowing the retriever to focus only on semantically relevant relations and thus reducing costly KG queries. Besides, ToG requires the final answer to be generated by the LLM, whereas VoG directly obtains the answer from the revised reasoning plan without additional generation. While PoG similarly benefits from similar adaptive search and yields overall similar efficiency to ours, it incorporates additional steps and modules that introduce extra token usage and memory overhead (see lines 423-424).
>
> We have incorporated the inference time into our efficiency analysis (see Section 4.4).

---

> > ### Comment · Reviewer_rU7C · 2025-11-22
> >
> > Thanks for your rebuttal. I think you addressed my concerns, and I will keep my positive score.

---

> > > ### Author Response · Authors · 2025-11-23
> > >
> > > Thank you very much for your careful review and positive feedback. We are pleased that our rebuttal has resolved your concerns.

---

### Official Review · Reviewer_4vgR · 2025-10-31

**Soundness:** 2
**Presentation:** 2
**Contribution:** 2
**Rating:** 4
**Confidence:** 5

**Summary:**

This paper proposes a novel Verify-on-Graph (VoG) framework, which supports dynamic and context-aware LLM reasoning over KG through iterative retrieval, verification, and adaptive refinement. To be specific, the proposed method firstly employs a plan agent to generate reasoning chains, serving as guidance for multi-hop retrieval process. Then, stepwise verification and adaptive refinement is adopted to detect reasoning inconsistencies and make sure the correctness of subsequent reasoning steps. Finally, a confidence-based reward is designed to capture uncertain information for revision. Extensive experimental results demonstrate the effectiveness of the proposed method.

**Strengths:**

1.	This paper is well-organized and easy to follow.
2.	This paper presents a VoG framework consisting of three specialized LLM agents for retrieval, verification, and revision, which address the challenges of inflexible reasoning and limited utilization of information.
3.	This paper provides the source code to ensure the reproducibility of the proposed method.

**Weaknesses:**

1.	The figures could be further refined to enhance readability. In particular, the font size in Figures 1 and 3 is quite small, and Figure 7 appears to have low resolution.
2.	The paper may lack some baseline methods for comparison, such as GNN-RAG [1], SubgraphRAG [2].
3.	The core idea may not be highly novel, since the retrieval–plan–verify pipeline has been adopted in prior studies.

[1] Mavromatis, Costas, and George Karypis. "Gnn-rag: Graph neural retrieval for large language model reasoning." arXiv preprint arXiv:2405.20139 (2024).
[2] Li, Mufei, Siqi Miao, and Pan Li. "Simple is Effective: The Roles of Graphs and Large Language Models in Knowledge-Graph-Based Retrieval-Augmented Generation." The Thirteenth International Conference on Learning Representations.

**Questions:**

Please refer to Section Weaknesses.

---

> ### Author Response · Authors · 2025-11-21
>
> We appreciate the reviewer’s thoughtful feedback. We are glad that the reviewer found our paper well-organized and recognized how VoG addresses existing challenges in the field. We address the reviewer’s comments as below：
>
> > **W1:** The figures could be further refined to enhance readability. In particular, the font size in Figures 1 and 3 is quite small, and Figure 7 appears to have low resolution.
>
> **A1:** We thank the reviewer for this helpful observation. We have refined the figures in our manuscript to enhance readability, including increasing the font size in Figures 1 and 3 and ensuring high resolution for all figures in the appendix.
>
> >  **W2:** The paper may lack some baseline methods for comparison, such as GNN-RAG [1], SubgraphRAG [2].
>
> **A2:** We appreciate this valuable suggestion. Following your suggestion, we have included comparisons with an additional class of baselines combining *finetuned retriever+LLM* in our updated version, such as GNN-RAG [1] and SubgraphRAG [2] (see Section 4 and Appendix D), as shown below:
>
> | **Model**                     | **CWQ Hits** | **CWQ F1** | **WebQSP Hits** | **WebQSP F1** |
> | ----------------------------- | ------------ | ---------- | --------------- | ------------- |
> | GNN‑RAG+RA (Fine-tuned LLAMA) | 68.7         | 60.4       | **90.7**        | **73.5**      |
> | GNN‑RAG (GPT‑3.5)             | 64.1         | -          | 85.3            | -             |
> | SubgraphRAG (GPT‑3.5)         | 56.3         | 49.1       | 83.1            | 69.2          |
> | VoG (GPT‑3.5)                 | 64.7         | 56.2       | 83.2            | 69.1          |
> | VoG (GPT‑4)                   | **77.6**     | **67.5**   | 88.7            | 73.2          |
>
> Note that these baselines are resource-consuming, which require both finetuned retrievers and additional LLM calls. In contrast, our VoG is training-free and of high generability, showing a comparable or even better performance.
>
> > **W3:** The core idea may not be highly novel, since the retrieval–plan–verify pipeline has been adopted in prior studies.
>
> **A3:** We thank the reviewer for raising this insightful concern. While a high-level retrieval–plan–verify structure has appeared in prior work for other tasks, VoG additionally introduces a stepwise revision module that **not only tackles the remaining challenges but also works synergistically with the other modules, collectively forming a feedback loop, as discussed below.**
>
> Specifically, we are aware that many works rely on retrieving external knowledge to verify LLM’s internal hallucinations. However, under long reasoning chains, they still suffer from challenges such as error propagation and the underutilization of retrieval feedback (see lines 82-87 and 93-98). Rather than following a sequential retrieve-then-verify execution, VoG enables dynamic revision of the reasoning plan, allowing errors in intermediate steps to be corrected timely. To effectively utilize retrieval feedback, VoG incorporates an MAB-based context selector that facilitates adaptive revision at each reasoning step.
>
> Besides, the dynamic feedback loop, where planning, retrieval, and verification continuously inform and refine each other, further distinguishes VoG from existing methods. In each step, the plan guides retrieval by providing semantically targeted relations based on a globally coherent full-chain plan. The retrieved KG evidence, in turn, not only corrects factual errors at local steps but also improves subsequent retrieval through context-aware revision, closing the loop and enabling the reasoning process to refine through each iteration. We have also refined Figure 1 to better illustrate this closed-loop interaction.
>
> We also provide the detailed methodological differences and compare VoG with recent KG-enhanced LLM reasoning methods in Sec. 5.2. To our knowledge, VoG is the first work that incorporates the explicit stepwise verification and adaptive revision into the KG reasoning process. If there are specific prior studies you believe closely match VoG’s formulation, we are willing to incorporate a targeted discussion.
>
> [1] Mavromatis, Costas, and George Karypis. "Gnn-rag: Graph neural retrieval for large language model reasoning." arXiv preprint arXiv:2405.20139 (2024).
>
> [2] Li, Mufei, Siqi Miao, and Pan Li. "Simple is Effective: The Roles of Graphs and Large Language Models in Knowledge-Graph-Based Retrieval-Augmented Generation." The Thirteenth International Conference on Learning Representations.

---

### Official Review · Reviewer_cbkJ · 2025-10-31

**Soundness:** 2
**Presentation:** 3
**Contribution:** 2
**Rating:** 4
**Confidence:** 5

**Summary:**

This paper proposes Verify-on-Graph (VoG), a model-agnostic framework that enhances reasoning reliability of LLMs on knowledge-intensive tasks by couping iterative retrieval, stepwise verification and revision, based on knowledge triplets retrieved from knowledge graphs. Unlike existing KG-augmented reasoning systems that follow fixed reasoning plans, VoG introduces (1) Stepwise KG verification that detects factual inconsistencies at each reasoning step. (2) Plan revision via multi-armed bandit context selection that adaptively decides which contextual scope to use for revising reasoning. Experiments on KG reasoning benchmarks show consistent improvements across backbone models of different sizes, compared to the strong agentic baselines with reduced token cost.

**Strengths:**

1. Novel verification framework for reasoning. The proposed step verification has the potential of mitigating error/hallucination propagation and improve faithful reasoning.

2. Adaptive context selection.  The idea of using UCB to balance the information window in the decision making process is innovative, and has the potential to be extended to other long-reasoning tasks as well.

3. Strong empirical results. Experiments show consistent gains across three benchmarks and multiple LLM sizes, which validate the approach's generality.

**Weaknesses:**

Some key components, such as reward design details for UCB are located in the Appendix. They should stand out in the body of the methodology. See Questions for other technical weakness.

**Questions:**

1. What is the motivation to revise the reasoning plan given the retrieved knowledge triplets? Why does the factual knowledge retrieved play a role in evaluating(verify) the reasoning sub-step? A more natural way could be to populate the retrieved knowledge when it is not sufficient to solve the reasoning sub-step.

2. There is no guarantee that the prompted based verification agent would return a faithful and correct response. In Table 2, the authors conducted ablation studies to show the effect of verify/revision. However, it is not clear whether the improvement is due to the verifier, or it's a effect of parallel thinking introduced by the revision itself. A more rigorous way is to test on a dataset that has ground truth reasoning([1] for example) steps and show these models can indeed distinguish the correct reasonings from the incorrect.

3. What is the benefit of having a plan first and conduct step-with verify and revise, instead of iteratively reasoning and retrieve as in [2]?

4.  VoG retrieves relation and entity based on some semantic similarity score, which is similar to the retrieval approach introduced in [3], it is beneficial to include a discussion or comparison on that.

5. Why the semantic similarity between the "predicted observation" and the input question can serve as a quality measure of reasoning, as introduced in Appendix B.2. Not every reasoning step needs to share similar semantic meaning with the question.

6. All experiments in Table 1 are conducted on Freebase knowledge graph, it remains unclear whether VoG generalizes to other domains, like scientific or biomedical KGs (such as UMLS).

[1] MINT-CoT: Enabling Interleaved Visual Tokens in Mathematical Chain-of-Thought Reasoning

[2] Search-R1: Training LLMs to Reason and Leverage Search Engines with Reinforcement Learning

[3] GIVE: Structured Reasoning of Large Language Models with Knowledge Graph Inspired Veracity Extrapolation

---

> ### Author Response · Authors · 2025-11-21
>
> We appreciate the reviewer’s comprehensive feedback. We are encouraged that the reviewer found our work innovative and has generalizable value. We address the reviewer’s comments below.
>
> > **W1:** Some key components, such as reward design details for UCB, are located in the Appendix. They should stand out in the body of the methodology.
>
> Thanks for pointing this out. Our updated manuscript has addressed this issue and has moved the key reward design details from the Appendix into the main text accordingly (see Section 3.4.2).
>
> > **Q1+Q3:** What is the benefit of having a plan first and conducting stepwise verification and revision, instead of iteratively reasoning and retrieving as in Search-R1? What is the motivation to revise the reasoning plan given the retrieved knowledge triplets? Why does the factual knowledge retrieved play a role in evaluating(verify) the reasoning sub-step? A more natural way could be to populate the retrieved knowledge when it is not sufficient to solve the reasoning sub-step.
>
> **A1+A3:** Thank you for raising the thoughtful concerns regarding (1) the motivation for having a plan first and then iteratively verifying and revising it using retrieved factual knowledge, and (2) pointing out the possible alternatives such as Search-R1 (iterative reason-and-retrieve paradigm) and populating missing knowledge whenever a sub-step appears insufficient. **We would like to clarify that our design choices are task-driven, and the differences from these alternatives outline the rationale behind our approach. The answers to your questions further made our motivation more explicit.** The details are as follows:
>
> First, we would like to clarify that the benefit of introducing an initial plan is not only theoretically coherent with the characteristics of multi-hop KGQA task, but also empirically validated through comparisons with other baselines. Specifically, multi-hop KGQA inherently depends on following a coherent graph path, whereas text-based RAG (like Search-R1[1]) can rely on locally iterating between reasoning and retrieval without maintaining a structured trajectory. In settings where traversing a complete and coherent KG path is necessary rather than a single piece of knowledge as noted in [2], the initial plan serves both as a global-level roadmap for retrieval and as contextual memory for maintaining coherence across long reasoning chains. In implementation, this design substantially improves KG-retrieval efficiency and downstream accuracy compared with purely iterative retrieval approaches such as ToG (see Table 1 and our updated Section 4.4). Moreover, the plan-guided retrieval further enables adaptive control over depth and breadth, which prevents irrelevant exploration and early termination of the reasoning process (see 159–161 and Appendix A.2).
>
> Regarding whether additional knowledge can be populated whenever a sub-step is insufficient, we would like to clarify that identifying such insufficiency from the LLM’s internal belief is fragile in our knowledge-intensive setting. This aligns with our concern that, although the initial plan provides useful global guidance, it is also generated purely from the LLM’s parametric knowledge and is thus vulnerable to factual inaccuracies especially when handling up-to-date or domain-specific information (see lines 35–37). This motivates us to place sub-step verification outside the LLM’s internal belief, which in turn lets the factual knowledge retrieved from the KG directly verify whether the predicted sub-step is correct.
>
> Together, this coherent plan-retrieve-revise loop allows VoG to fully exploit LLM’s own reasoning ability while simultaneously leveraging the structured, trustworthy knowledge from KG. Building on the above clarifications that further support our motivation, we have updated the manuscript and added a discussion of Search-R1[1] (see Section 5.1) and other natural alternatives.

---

> > ### Author Response · Authors · 2025-11-21
> >
> > > **Q2:** There is no guarantee that the prompted verification agent would return a faithful and correct response. In Table 2, the authors conducted ablation studies to show the effect of verify/revision. However, it is not clear whether the improvement is due to the verifier, or it's a effect of parallel thinking introduced by the revision itself. A more rigorous way is to test on a dataset that has ground truth reasoning (e.g., [3]) steps and show these models can indeed distinguish the correct reasoning from the incorrect.
> >
> > **A2:** We appreciate the reviewer’s constructive concern regarding the reliability of the verifier and the need for step-level evaluation, both of which we also regard as important and have discussed in our initial submission. To further address this, **we provide further clarification and strengthen the analysis in our original manuscript with two additional targeted evaluations**.
> >
> > First, we agree with the reviewer that relying solely on a prompted LLM cannot guarantee faithful or fully correct verification. This is exactly why we incorporate both a prompted LLM verifier and a PLM-based semantic consistency checker (Section 3.3 and Appendix E.1) to better guarantee faithfulness. As described in Section 3.3 and Equation (1), our verification agent is designed to identify factual inconsistencies between the predicted observation and the retrieved KG evidence, rather than to assess the full correctness of a reasoning path which has ground-truth supervision at the final answer level. As a result, there is no task-provided step-level ground truth to support a binary accuracy or error-rate evaluation for verifier alone. Accordingly, we include an end-to-end performance comparison between a PLM-only verifier and our verifier setup to isoloate its overall effect:
> >
> > | **Setting**          | **CWQ Accuracy (%)** | **WebQSP Accuracy (%)** |
> > | -------------------- | -------------------- | ----------------------- |
> > | PLM-based Verifier   | 62.5                 | 80.4                    |
> > | VoG's *Verify Agent* | 64.7                 | 83.2                    |
> >
> > Besides, we would like to clarify that the improvement observed in Table 2 results from the coupled verification and revision mechanism, since revision is only triggered by signals given by *verify agent* (see lines 201-202). Regarding your concern about step-level revision evaluation based on ground-truth, our initial submission also provided an analysis that uses the ground-truth final answer as the criterion, reporting the revision success rates of different context strategies (see Appendix E.3). To better reflect step-level correctness as suggested in [3], we additionally evaluate whether the model can distinguish and correct incorrect intermediate reasoning steps using ground-truth SPARQL paths. In this setup, an LLM-based evaluator (GPT-3.5) judges each revised step as (i) CORRECT, fully consistent with the gold reasoning path, (ii) IMPROVED, an improvement over an originally incorrect step, or (iii) INCORRECT. The proportions are shown below:
> >
> > **CWQ:**
> >
> > | **Metric**         | **Proportion** |
> > | ------------------ | -------------- |
> > | CORRECT + IMPROVED | 85.21%         |
> > | CORRECT only       | 44.36%         |
> >
> > **WebQSP:**
> >
> > | **Metric**         | **Proportion** |
> > | ------------------ | -------------- |
> > | CORRECT + IMPROVED | 99.50%         |
> > | CORRECT only       | 83.25%         |
> >
> > We have further revised our draft accordingly to improve the clarity of how VoG’s verification–revision mechanism contributes to faithful reasoning with additional validation (see appendix E.3).

---

> > > ### Author Response · Authors · 2025-11-21
> > >
> > > > **Q4:** VoG retrieves relation and entity based on some semantic similarity score, which is similar to the retrieval approach introduced in [4]. It is beneficial to include a discussion or comparison on that.
> > >
> > > **A4:** We appreciate the valuable suggestion to discuss the relation between VoG’s and the approach in GIVE [4]. We address this concern through **a distinction between VoG and GIVE,** and a **comparison experiment under GIVE’s sparse-KG setting**. Regarding the semantic scoring, we note that it is a standard candidate-pruning module shared across many LLM+KG reasoning frameworks, and the relevant comparisons are already reflected in our main results (see Table 1, Sec 4.2).
> > >
> > > Specifically, GIVE performs semantic clustering only within 1-hop, which is appropriate for sparse and domain-specific KGs but not for path-based multi-hop reasoning. As noted by the authors of GIVE, their method is not intended to generalize to dense KGs such as Freebase. In contrast, VoG uses semantic scoring for local-hop pruning within a structured path-search process, allowing it to operate on large and dense KGs. In addition, we conducted a small-scale comparison evaluation under the sparse-KG conditions used in GIVE [4] as shown below:
> > >
> > > | **Dataset** | **Method** | **Accuracy (%)** | **Avg Tokens** | **Avg Time (s)** |
> > > | ----------- | ---------- | ---------------- | -------------- | ---------------- |
> > > | PubMedQA    | GIVE       | 53.6             | 14,701.5       | 33.6             |
> > > | PubMedQA    | ToG        | 17.6             | 12,805.3       | 15.9             |
> > > | PubMedQA    | VoG        | 45.3             | 1,653.2        | 12.1             |
> > > | BioASQ      | GIVE       | 88.2             | 8,050.3        | 15.3             |
> > > | BioASQ      | ToG        | 18.0             | 7,070.2        | 10.3             |
> > > | BioASQ      | VoG        | 93.5             | 1,862.6        | 14.1             |
> > >
> > > While ToG fails consistently due to the lack of candidate relations on sparse KG (a UMLS-derived KG with 135 nodes and 5,877 edges), VoG remains competitive by leveraging the LLM-based initial planning step to compensate using internal knowledge, achieving comparable performance with substantially fewer tokens. These results confirm that VoG remains efficient and effective across sparse-KG domains as well.
> > >
> > > We have revised the draft accordingly to incorporate this discussion and comparison (see Appendix G).
> > >
> > > > **Q5:** Why does the semantic similarity between the "predicted observation" and the input question serve as a quality measure of reasoning, as introduced in Appendix B.2? Not every reasoning step needs to share similar semantic meaning with the question.
> > >
> > > **A5:** Thanks for the insightful question on our detailed reward design. We address this concern by **clarifying the intended motivation** of this measure and **providing empirical evidence** showing its contribution to stabilizing multi-hop reasoning.
> > >
> > > Specifically, we would like to clarify that in multi-hop QA, each intermediate hop corresponds to progressing along the latent relation path that leads to the final answer. In this setting, each step needs to share a similar semantic meaning with the query’s underlying relational intent. (e.g., “which film was directed by the author of X?” corresponding to the path “X —written_by→ author —directed→ film”). In datasets such as CWQ, this alignment is explicit because the questions are constructed as a paraphrase of the underlying reasoning path [5]. This motivates us to use the similarity score as a lightweight indicator for assessing whether the observation in revised plan remains aligned with the intended trajectory, complementing our *validation* metric of factual correctness.
> > >
> > > Empirically, our ablation studies (see Table 11, Appendix E.6) further support the usefulness of this metric. Removing this *Quality* signal (semantic-similarity based) consistently lowers the final-answer accuracy obtained in our setup. If you have any specific suggestions for additional step-quality measures, we will incorporate an ablation or discussion as soon as possible.

---

> > > > ### Author Response · Authors · 2025-11-21
> > > >
> > > > > **Q6:** All experiments in Table 1 are conducted on Freebase knowledge graph, and it remains unclear whether VoG generalizes to other domains, like scientific or biomedical KGs (such as UMLS).
> > > >
> > > > **A6:** We demonstrate VoG's generalization to other KGs by supplementing our main results with **two further experiments that evaluate VoG beyond Freebase and toward domain-specialized KGs**.
> > > >
> > > > - Generalization beyond Freebase
> > > >
> > > >   We transferred VoG from Freebase to Wikidata and evaluated it on the QALD-10 [6] dataset. As Wikidata remains an open-domain KG , we only replaced the KG-specific retrieval interface without requiring prompt redesign. VoG achieves strong performance compared to previous training-based and training-free methods:
> > > >
> > > >   | **Model**   | **Method**    | **Accuracy(%)** |
> > > >   | ----------- | ------------- | --------------- |
> > > >   | Fine-Tuned  | SPARQL-QA [6] | 45.4            |
> > > >   | ChatGPT 3.5 | ToG           | 50.2            |
> > > >   | ChatGPT 3.5 | VoG           | 57.4            |
> > > >
> > > > - Generalization to medical domains
> > > >
> > > >   Currently, achieving generalization across KGs remains an open challenge among prior methods, including most of our baselines such as PoG and RoG, partly because most domain-specific KGs (e.g., biomedical or scientific KGs) lack high-quality multi-hop QA datasets and existing resources are often yes/no style or have long free-form answers. Notably, PoG also received similar requests for cross-KG evaluation during peer review but was unable to include such experiments due to the same dataset limitations. Here we try our best to construct a small biomedical multi-hop QA test set using Hetionet [7], a UMLS-derived KG with 47,031 nodes and 2,250,197 edges. We extract multi-hop relational paths, filter them for coherence, and use LLMs to rewrite them into 200 naturally phrased biomedical questions (e.g., "Which genes interact with genes that are downregulated by Dacarbazine?"). Given that biomedical queries exhibit far less contextual variability than open-domain ones, we applied the local revision strategy for this pilot study. The results are shown below：
> > > >
> > > >   | **Setting**                   | **Accuracy (%)** |
> > > >   | ----------------------------- | ---------------- |
> > > >   | Initial GPT Answer (baseline) | 1.0              |
> > > >   | VoG Final Answer (ours)       | 19.5             |
> > > >   | Gain                          | +18.5            |
> > > >
> > > > VoG’s cross-KG transferability primarily comes from two design principles:
> > > >
> > > > - **Training-free pipeline**: All components in VoG are training-free. Retrieval relies on pretrained text embeddings, which capture transferable semantics across heterogeneous KGs, enabling effective pruning even without domain-specific tuning.
> > > > - **LLM-driven planning + verify–revise loop**: LLMs possess inherent **cross-domain reasoning ability**, which our planning module leverages. The verify–revise process then stabilizes the reasoning trajectory with KG evidence, even when domain knowledge is sparse or incomplete.
> > > >
> > > > We regard this as a preliminary exploration and have incorporated a brief discussion in our updated manuscript (see Appendix G). In the future, we will explore constructing higher-quality datasets across diverse domain KGs and further expand this direction with a more systematic evaluation.
> > > >
> > > > [1] Jin, Bowen, et al. "Search-r1: Training llms to reason and leverage search engines with reinforcement learning." *arXiv preprint arXiv:2503.09516* (2025).
> > > >
> > > > [2] Song, Yiqing , et al. "Advancements in Complex Knowledge Graph Question Answering: A Survey." *Electronics* 12.21(2023):16.
> > > >
> > > > [3] Chen, Xinyan, et al. "MINT-CoT: Enabling Interleaved Visual Tokens in Mathematical Chain-of-Thought Reasoning." *arXiv preprint arXiv:2506.05331* (2025).
> > > >
> > > > [4] He, Jiashu, et al. "Give: Structured reasoning of large language models with knowledge graph inspired veracity extrapolation." *arXiv preprint arXiv:2410.08475* (2024).
> > > >
> > > > [5] Talmor, Alon, and Jonathan Berant. "The web as a knowledge-base for answering complex questions." *arXiv preprint arXiv:1803.06643* (2018).
> > > >
> > > > [6] Borroto, Manuel, et al. "SPARQL-QA enters the QALD challenge." *Proceedings of the 7th Natural Language Interfaces for the Web of Data (NLIWoD) co-located with the 19th European Semantic Web Conference, Hersonissos, Greece*. Vol. 3196. 2022.
> > > >
> > > > [7] Himmelstein, Daniel Scott, et al. "Systematic integration of biomedical knowledge prioritizes drugs for repurposing." *elife* 6 (2017): e26726.

---

> > > > > ### Comment · Reviewer_cbkJ · 2025-11-22
> > > > >
> > > > > Thank you for your rebuttal. Your response resolved my concerns. I will raise my rating to 6.

---

> > > > > > ### Author Response · Authors · 2025-11-23
> > > > > >
> > > > > > We are glad to know that your concerns are addressed. We greatly appreciate your efforts in reviewing our paper and providing comprehensive feedback.

---

### Author Response · Authors · 2025-11-21

We thank all the reviewers for their valuable feedback, which has helped us clarify key points and improve the overall presentation of the paper.

- We have made several revisions to our manuscript and highlighted changes corresponding to each reviewer using color annotations for clarity: changes addressing reviewer cbkJ are marked in blue, reviewer 4vgR in orange, reviewer rU7C in green, and reviewer YG4Q in purple. Edits that respond to concerns shared by reviewer cbkJ and YG4Q are marked in dark blue. Several figures have also been refined following the suggestions of reviewer 4vgR.
- In addition, we expanded the manuscript by incorporating new experiments and enhancing multiple analyses, including (i) generalization to KGs beyond Freebase (cbkJ, YG4Q); (ii) a trade-off analysis between complexity and performance (YG4Q); (iii) improved efficiency study, ablation analysis, and robustness study (rU7C, cbkJ, YG4Q); and (iv) additional comparisons and discussions with further baselines (cbkJ, 4vgR).

Below, we respond to each question in its respective review.

---

### Author Response · Authors · 2025-12-02
**Rebuttal Summary**

We sincerely thank the reviewers and the ACs for their time and efforts under the challenging circumstances of this year. We would like to clarify that the score for our submission was updated **from 8644 to 8664** by reviewer cbkJ at **21:31 UTC on Nov 22**, which was **before the broader public disclosure of the OpenReview bug**. We strictly followed the ICLR Code of Conduct throughout the process and never used any API, tool, or method related to the leaked information.

The major improvements made in response to the reviewers’ concerns are summarized below:

- **Clarified methodological contributions and motivation.** We strengthened the explanation of VoG’s design choices (reviewer cbkJ) and clarified its distinctions from existing retrieval–plan–verify paradigms (reviewer 4vgR).
- **Expanded empirical evaluation.** We demonstrate VoG’s generalization ability with additional cross-KG experiments (reviewers cbkJ and YG4Q) and strengthened the experimental analysis with additional efficiency evaluations (reviewer rU7C), robustness studies (reviewer YG4Q), trade-off analysis (reviewer YG4Q), verification analyses (reviewer cbkJ), and extended comparisons with relevant baselines such as GIVE (reviewer cbkJ), GNN-RAG, and SubgraphRAG (reviewer 4vgR).
- **Improved presentation and clarity.** We refined multiple figures (reviewer 4vgR), adjusted the placement of key methodological components (reviewer cbkJ), and revised the explanation of specific cases (reviewer YG4Q) to improve clarity and readability.

Together, we believe that the above clarifications, analyses, and revisions comprehensively address the reviewers’ key concerns. Although reviewer 4vgR could not respond further because the discussion period was prematurely terminated, reviewers cbkJ, rU7C, and YG4Q had already acknowledged that their concerns were resolved before **Nov 23 03:36 UTC**. We respectfully hope that the AC will consider the valid score update, along with our complete rebuttal of all raised issues, when making the final decision.

---

### Meta-Review · Area_Chair_a8CE · 2026-01-12

**Summary:**

This paper introduces Verify-on-Graph (VoG), a model-agnostic framework that improves the reliability of LLM reasoning on knowledge-intensive tasks by integrating iterative retrieval, stepwise verification, and adaptive plan revision using knowledge graph (KG) triplets. Unlike static KG-augmented methods, VoG dynamically verifies factual consistency at each reasoning step and employs a multi-armed bandit strategy to adaptively select the most informative contextual scope for revision. A confidence-based reward further guides refinement under uncertainty. Experiments across multiple backbone models and KG reasoning benchmarks show consistent performance gains over strong agentic baselines, with lower token usage, demonstrating VoG’s efficiency and effectiveness in enabling accurate, context-aware, and self-correcting reasoning. The proposed idea is interesting and the paper is well written.

**Reviewer Scores:**

NA

---

### Decision · Program_Chairs · 2026-01-26

Accept (Poster)